# Intracellular interleukin-32γ mediates antiviral activity of cytokines against hepatitis B virus

Doo Hyun Kim[1], Eun-Sook Park[1], Ah Ram Lee[1], Soree Park[1], Yong Kwang Park[1], Sung Hyun Ahn[1], Hong Seok Kang[1], Ju Hee Won[1], Yea Na Ha[1], ByeongJune Jae[1], Dong-Sik Kim [2], Woo-Chang Chung[3], Moon Jung Song[3], Kee-Hwan Kim[4], Seung Hwa Park[5], Soo-Hyun Kim[6] & Kyun-Hwan Kim[1,7]

Cytokines are involved in early host defense against pathogen infections. In particular, tumor necrosis factor (TNF) and interferon-gamma (IFN-γ) have critical functions in non-cytopathic elimination of hepatitis B virus (HBV) in hepatocytes. However, the molecular mechanisms and mediator molecules are largely unknown. Here we show that interleukin-32 (IL-32) is induced by TNF and IFN-γ in hepatocytes, and inhibits the replication of HBV by acting intracellularly to suppress HBV transcription and replication. The gamma isoform of IL-32 (IL-32γ) inhibits viral enhancer activities by downregulating liver-enriched transcription factors. Our data are validated in both an in vivo HBV mouse model and primary human hepatocytes. This study thus suggests that IL-32γ functions as intracellular effector in hepatocytes for suppressing HBV replication to implicate a possible mechanism of non-cytopathic viral clearance.

[1] Department of Pharmacology and Center for Cancer Research and Diagnostic Medicine, IBST, School of Medicine, Konkuk University, Seoul 05029, Republic of Korea. [2] Division of HBP Surgery and Liver Transplantation, Department of Surgery, Korea University College of Medicine, Seoul 02841, Republic of Korea. [3] Virus-Host Interactions Laboratory, Division of Biotechnology, Department of Biosystems and Biotechnology, College of Life Sciences and Biotechnology, Korea University, Seoul 02841, Republic of Korea. [4] Department of Surgery, Uijeongbu St. Mary's Hospital, Catholic Central Laboratory of Surgery, College of Medicine, The Catholic University of Korea, Seoul 11765, Republic of Korea. [5] Department of Anatomy, School of Medicine, Konkuk University, Seoul 05029, Republic of Korea. [6] Laboratory of Cytokine Immunology, Veterinary School, Konkuk University, Seoul 05029, Republic of Korea. [7] KU Open Innovation Center, Research Institute of Medical Sciences, Konkuk University, Seoul 05029, Republic of Korea. These authors contributed equally: Doo Hyun Kim, Eun-Sook Park. Correspondence and requests for materials should be addressed to K.-H.K. (email: khkim10@kku.ac.kr)

nfection with hepatitis B virus (HBV) is responsible for global public health problems including chronic hepatitis B, liver cirrhosis, and hepatocellular carcinoma. HBV clearance during acute HBV infection is mainly mediated by antiviral cytokines secreted by cytotoxic T lymphocytes without damage to infected hepatocytes, a phenomenon dubbed non-cytopathic clearance[1–3]. Various cytokines inhibit HBV gene expression and replication through diverse mechanisms in vitro and in vivo[4,5]. Tumor necrosis factor-alpha (TNF-α) reduces HBV RNA production and capsid stability[1–3,6]. Interferon-gamma (IFN-γ) eliminates pregenomic RNA-containing capsids in mouse hepatocytes[5]. Interleukin (IL)-4 inhibits HBV transcription by downregulating CCAAT/enhancer-binding protein (C/EBP)[7].

TNF-α and IFN-γ are representative antiviral cytokines that suppress HBV in a non-cytolytic manner[1,3,8]. Recently, T cell-derived TNF-α and IFN-γ were found to reduce HBV covalently closed circular DNA (cccDNA) in hepatocytes by inducing APOBEC3 deaminases[9]. We also reported that P22-FLIP[10] and hepatocystin[11] are involved in TNF-α-mediated and IFN-γ-mediated suppression of HBV, respectively. However, the precise mechanism and downstream mediator molecules involved in HBV suppression by TNF-α and IFN-γ have not been clearly elucidated.

IL-32 was discovered as a proinflammatory cytokine that is secreted from natural killer (NK) cells and activates mitogen-activated protein kinases (MAPKs) and NF-κB signaling pathways[12,13]. It is produced by various epithelial and immune cells such as T lymphocytes, NK cells, and monocytes. In the pancreas, it is induced by cytokines[14]. The IL-32 gene is transcribed into six alternative splice variants; among them, IL-32γ is the most active form[15]. However, the biological function of IL-32 isoforms in specific tissues remains unclear.

Treatment with recombinant IL-32 suppresses replication of various viruses through autocrine or paracrine mechanisms (i.e., IL-32 acts as a typical cytokine); the viruses suppressed by IL-32 include vesicular stomatitis virus (VSV)[16,17], human immunodeficiency virus (HIV)[18,19], and influenza virus[20–22]. Interestingly, treatment of peripheral blood mononuclear cells with recombinant IL-32-induced IFN-λ1, which in turn inhibited HBV replication in hepatocytes[23]. However, the direct antiviral action of IL-32 in hepatocytes has not been demonstrated.

Transcription of HBV cccDNA in the nucleus requires liver-enriched transcription factors such as hepatocyte nuclear factors (HNFs) and C/EBP, which bind to HBV enhancer I (EnhI) and II (EnhII) regions[24,25]. The expression of these factors is regulated mainly by MAPK signaling pathways[26,27]. Here we show that TNF-α and IFN-γ synergistically induce IL-32, which suppresses HBV by downregulating liver-enriched transcription factors in a non-cytokine-like manner. Our novel finding suggests a mechanism of cytokine-mediated non-cytopathic clearance of HBV.

## Results

### Cytokine-induced intracellular IL-32 inhibits HBV replication.
As HBV-specific CD8+ cytotoxic T cells suppress viral replication by releasing TNF-α and IFN-γ[4], we tested whether these antiviral cytokines induce the expression of IL-32. Treatment of Huh7 cells with either TNF-α or IFN-γ upregulated the IL-32 protein in a dose-dependent manner. Surprisingly, co-treatment with TNF-α and IFN-γ strongly induced endogenous IL-32 in a synergistic manner (Fig. 1a, left). Upon treatment with these cytokines, cells remained intact (Fig. 1a, right). Immunofluorescence analysis confirmed that the IL-32 protein is induced by the cytokines and revealed that it is mostly localized in the cytoplasm. Ectopic expression of IL-32 was used as a positive control (Fig. 1b).

To assess the anti-HBV effect of TNF-α and IFN-γ, Huh7 cells were transfected with HBV 1.2 and treated with these cytokines. As expected, Southern blot analysis showed that co-treatment with TNF-α and IFN-γ synergistically inhibited HBV replication (Fig. 1c, left). Under this condition, the expression of IL-32 was strongly induced without loss of cell viability (Fig. 1c, right). To test whether cytokine-induced IL-32 is involved in the anti-HBV effect of TNF-α and IFN-γ, we knocked down IL-32 in Huh7 and HepG2 cells and confirmed it is silencing by western blot analysis (Supplementary Fig. 1). In both cell lines, the knockdown restored HBV replication inhibited by cytokine treatment (Fig. 1d).

Because IL-32 was discovered as a cytokine secreted by NK cells[12,13], we tested whether the induced IL-32 exerts its anti-HBV effect through an autocrine or paracrine effect. To check this, Huh7 and HepG2 cells transfected with HBV 1.2mer were incubated with the recombinant human IL-32γ (rhIL-32γ). Unexpectedly, this treatment had no effect on HBV replication in both cell lines (Fig. 1e). The biological activity of rhIL-32γ was confirmed by its ability to induce IL-8 and TNF-α in human THP-1 and murine Raw 264.7 cells, which was reported previously[15] (Supplementary Fig. 2). These unexpected results prompted us to check whether cytokine-induced IL-32 is secreted or not. Surprisingly, it was mostly retained in the cells and was rarely detected in culture supernatants of Huh7 hepatoma cells (Fig. 1f). To check whether the intracellular retention of IL-32 is physiologically relevant, we treated PHHs and differentiated HepaRG cells with TNF-α and IFN-γ. The results clearly revealed that IL-32 was not secreted from hepatocytes (Fig. 1g, Top HepaRG; Bottom PHHs).

Taken together, these findings suggest that IL-32 mediates the anti-HBV activity of TNF-α and IFN-γ as a non-cytokine-like molecule in hepatocytes.

### Intracellular antiviral effect of IL-32 is specific to HBV.
To confirm that the IL-32γ-mediated inhibition of HBV replication is an intracellular event, we co-transfected Huh7 cells with HBV 1.2 and IL-32γ expression plasmids. Overexpression of IL-32 strongly inhibited HBV replication (by approximately 90%) with no evidence of cytotoxicity (Fig. 2a). As in the experiments described above, little IL-32γ was secreted when it was overexpressed (Fig. 2b). When IL-32γ was overexpressed in HepG2, L02, and Hep3B cells, it also strongly inhibited HBV replication (Supplementary Fig. 3a–d). To confirm the direct inhibitory effect of overexpressed IL-32γ on viral replication, we knocked down IL-32γ by using siRNA. Silencing of IL-32 completely restored HBV replication (Supplementary Fig. 3b).

In addition, we tested which isoform of IL-32 is most active in inhibiting HBV replication. Although all three IL-32 isoforms tested (IL-32α, IL-32β, and IL-32γ) inhibited HBV replication, IL-32γ had the most potent anti-HBV activity (Fig. 2c, d).

Next, we investigated whether the antiviral effect of IL-32γ is mediated by the induction of general antiviral pathways, such as innate immune responses, or is specific to HBV. We infected Huh7 cells with herpes simplex virus 1 (HSV-1_GFP) or influenza A virus (A/WSN/33_GFP) and assayed the effect of IL-32γ on viral growth. Ganciclovir (GCV) and a combination of IFN-α and IFN-γ were used as positive control antiviral agents for each virus. First, we found that HSV-1 is insensitive to a combination of TNF-α and IFN-γ, which inhibits HBV replication; however, influenza virus was sensitive to these cytokines (Fig. 2e, f). Analysis of GFP signals and viral titration showed that IL-32γ expression did not affect the replication of HSV-1 or influenza virus in Huh7 cells (Fig. 2e–h). Analysis of viral transcription, protein expression, and titration (Supplementary

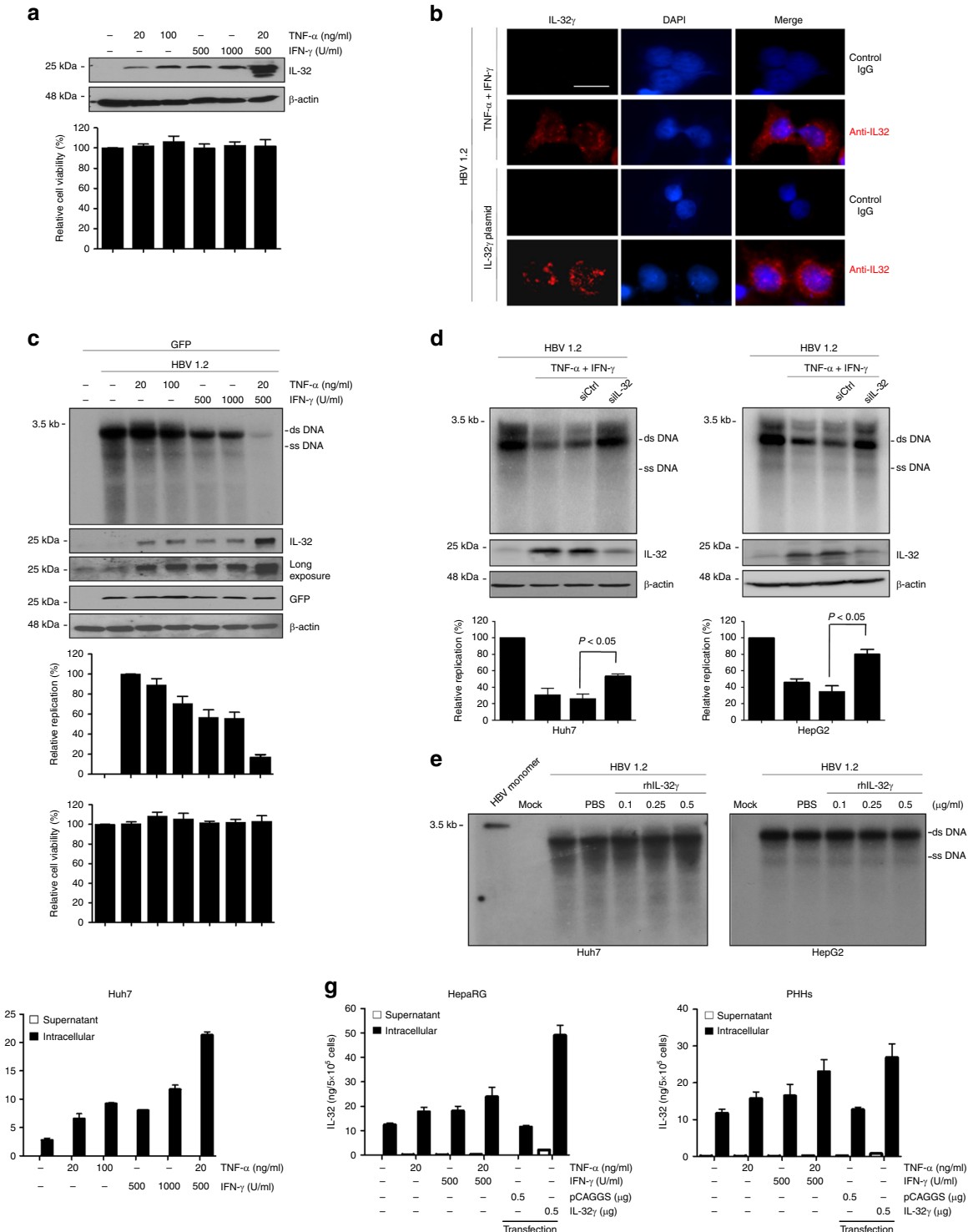

**Fig. 1** Induction of IL-32 is involved in cytokine-mediated inhibition of HBV. **a** Huh7 cells were treated with the indicated concentrations of TNF-α and IFN-γ for 48 h, and the IL-32 level was determined by western blot analysis. Relative cell viability was measured by the XTT method. **b** Expression of IL-32γ induced by cytokines or transient transfection was observed by immunofluorescence assay. Magnification, ×400; scale bar, 50 μm. **c** Huh7 cells were transfected with the HBV 1.2 and GFP plasmids and treated with TNF-α and IFN-γ. HBV replication was determined by Southern blotting. Expression levels of IL-32 and GFP (transfection control) were determined by western blotting. **d** HBV 1.2 and siRNAs (20 nM) were co-transfected into Huh7 or HepG2 cells. Next day, cytokines were added with fresh medium. HBV replication and IL-32 expression were determined by Southern and western blotting, respectively. $p < 0.05$ by Student's t-test. **e** Huh7 or HepG2 cells were transfected with the HBV 1.2 plasmid and treated with recombinant human IL-32γ (rhIL-32γ) for 3 days. HBV replication was determined by Southern blotting. **f** The levels of the IL-32 protein in culture medium and lysates of cytokine-treated Huh7 cells were determined by ELISA. The total amount of secreted (supernatant) or intracellular IL-32 in whole cells grown in a 12-well plate are shown. **g** The levels of the IL-32 protein in culture medium and lysates of cytokine-treated PHHs and differentiated HepaRG cells were determined by ELISA. Dashed bars represent the level of IL-32 obtained after transfection with pCAGGS (control) or IL-32 plasmids. Data (**a**, **c**, **d**, **f**, **g**) were obtained from three independent experiments (mean ± S.D.). $p < 0.05$ by Student's t-test

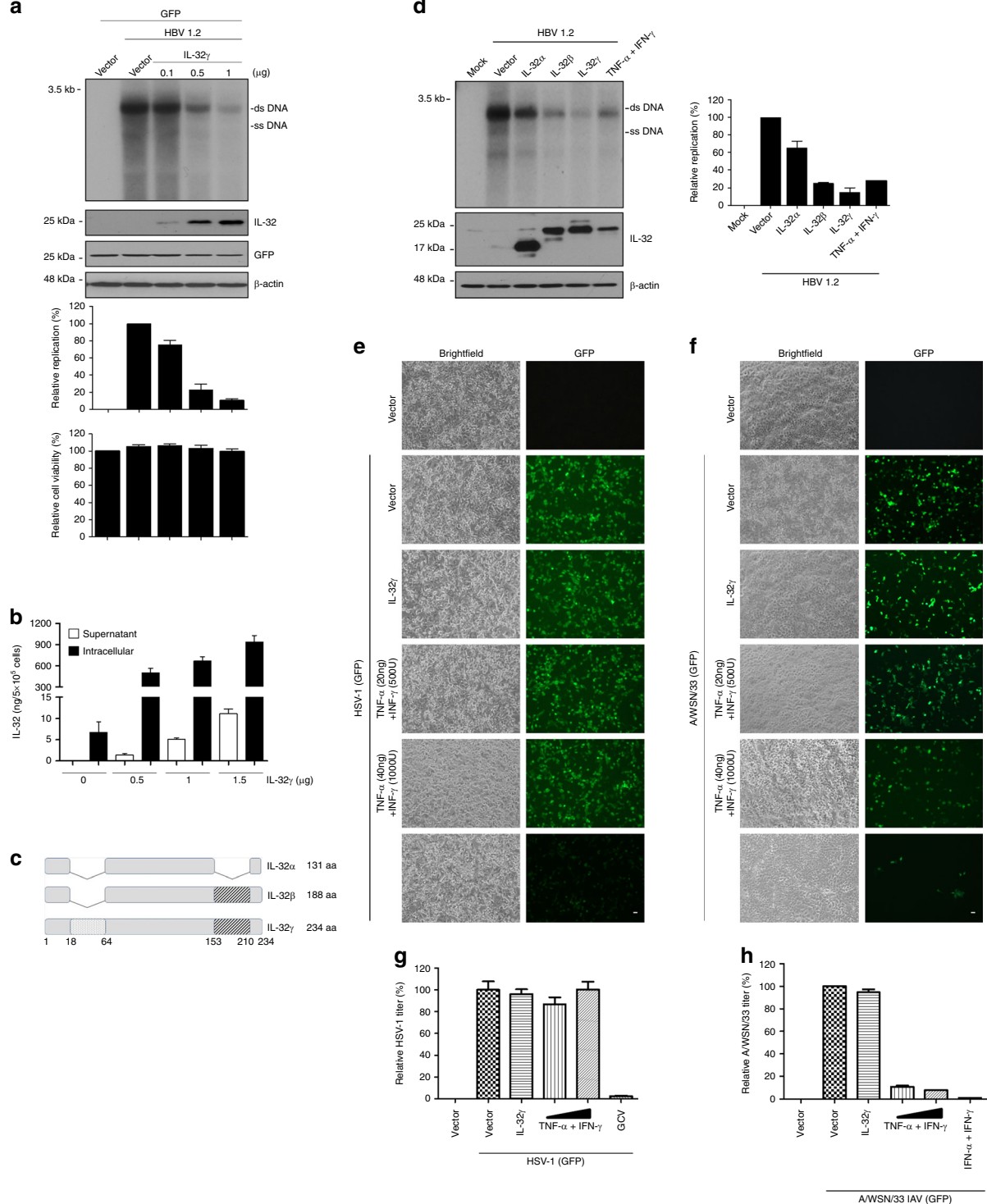

**Fig. 2** Inhibition of HBV replication by IL-32γ through an intracellular event and the effect of IL-32γ on other viruses. **a** Vectors for HBV 1.2 and IL-32γ were co-transfected into Huh7 cells. At 48 h post-transfection, viral replication, protein expression, and cell viability were analyzed by Southern blotting, western blotting, and XTT assay, respectively. **b** The levels of the IL-32 protein in culture medium and lysates of IL-32γ-transfected Huh7 cells were measured by ELISA. **c** Schematic illustration of IL-32 isoforms. Shaded boxes represent the N-terminal and C-terminal extensions in IL-32γ. **d** Vectors for HBV 1.2, IL-32α, IL-32β, or IL-32γ were transfected in Huh7 cells as indicated. At 48 h post-transfection, viral replication and protein expression were analyzed by Southern and western blotting, respectively. Cells treated with cytokines were used as a control. **e**–**h** IL-32 expression vector or empty vector was transfected into Huh7 cells. After 16 h, cells were infected with herpes simplex virus 1 (HSV-1_GFP) or influenza A virus (A/WSN/33_GFP) at 1 MOI. Cells were treated with the indicated concentrations cytokines (TNF-α and IFN-γ). Treatment with GCV (1 μg/mL) or IFNs (1000 U/mL IFN-α and 100 U/mL IFN-γ) was used as a positive control for HSV-1 and A/WSN/33, respectively. The GFP signals were monitored at 24 h post-infection. Magnification, ×100; scale bar, 100 μm. Data (**a**, **b**, **d**, **g**, **h**) were obtained from three independent experiments (mean ± S.D.)

Figs. 4 and 5) also indicated that IL-32γ exerts no effect on replication of HSV-1 or influenza virus. Indeed, ectopically expressed IL-32γ did not induce the antiviral cytokines and cytokine-induced antiviral genes in hepatocytes (Supplementary Fig. 6). These results suggest that IL-32γ-mediated inhibition of HBV replication is virus-specific in hepatocytes.

**IL-32γ suppresses HBV at transcriptional level**. To identify at which step of the HBV life cycle IL-32γ interferes, we first analyzed viral transcription using Northern blot analysis. Transfection with IL-32γ inhibited the expression of HBV mRNAs in a dose-dependent manner (Fig. 3a). Accordingly, the levels of viral proteins such as the surface and core proteins were remarkably reduced by IL-32γ (Fig. 3b). As the transcription of the HBV genome is mainly regulated by EnhI and by EnhII, which overlaps with the core promoter (EnhII/Cp), we determined which enhancer is targeted by IL-32γ. We generated reporter plasmids containing deletion mutants of the enhancer regions (Fig. 3c) and determined their activities in the absence or presence of IL-32γ. Reporter assay showed that EnhI had much higher activity than EnhII/Cp when IL-32γ was absent. The activities of all enhancer reporters were markedly decreased (by ~90%) by IL-32γ, suggesting that IL-32γ suppresses both EnhI and EnhII/Cp simultaneously (Fig. 3d).

As these results suggest that IL-32 inhibits HBV at the transcriptional level, we first determined the effect of IL-32 knockdown on cytokine-induced suppression of HBV by Northern blot analysis (Fig. 3e). The cytokine-induced decrease in HBV RNAs was significantly reverted by IL-32 knockdown in human hepatocytes (Fig. 3e). The steady-state level of HBV transcription was suppressed by both IL-32γ and IFN-γ in the nuclear and cytoplasmic compartments (Fig. 3f). Furthermore, nuclear run-on assay revealed that the rate of HBV transcription was decreased by IL-32γ and IFN-γ (Fig. 3g). Together with IL-32-mediated inhibition of enhancer activity (Fig. 3d), our data suggest that cytokines inhibit HBV at the transcriptional level by inducing IL-32, consistent with our and other previous studies which found that cytokines inhibit HBV at the transcriptional level in human hepatocytes[10,28,29].

As it is well known that IFN-γ inhibits HBV at the post-transcriptional level in a mouse model[30], we freshly prepared PMHs from mouse liver and analyzed the effect of cytokines on HBV replication and antigen expression. The level of viral RNA and antigen secretion were decreased by cytokines and IL-32γ in PMHs (Supplementary Fig. 7).

Next, we checked whether the SSB/La protein level is affected by cytokine treatment or IL-32γ expression because the stability of HBV RNA is reduced by cytokine-induced depletion of the La protein in a mouse model[31–33]. Indeed, cytokines destabilized the La protein in PMHs; however, they showed little effect in Huh7 human cells (Supplementary Fig. 7). Importantly, the level of the La protein was not affected by overexpression of IL-32γ in either cell type, suggesting that cytokine-induced IL-32 downregulates HBV RNA in a La-independent manner.

To further confirm the La-independence in human hepatocytes, we determined the levels of HBV RNAs after a knockdown of La expression (Supplementary Fig. 8). In Huh7 cells, La knockdown had no effect on the levels of HBV RNAs, whereas IL-32 strongly reduced them (Supplementary Figs. 8 and 9). However, LA knockdown in PMHs significantly decreased the levels of HBV RNAs and antigens (Supplementary Fig. 9). These data suggest that cytokines suppress HBV RNAs at the post-transcriptional level through a La-dependent pathway in PMHs but at the transcriptional level through a La-independent pathway in human hepatocytes.

**IL-32γ decreases HNFs expression**. HBV transcription is regulated by binding of liver-enriched transcription factors to enhancers and the core promoter[7,25,34]. As IL-32γ suppresses both EnhI and EnhII/Cp simultaneously, we hypothesized that the transcription factors that bind to both regions are targeted by IL-32γ. A literature search revealed that several ubiquitous and liver-enriched transcription factors bind to both EnhI and EnhII/Cp[27,35]. Among those, we focused on the major liver-enriched transcription factors known to be involved in HBV transcription including C/EBP, HNF1, HNF3, and HNF4 (Fig. 4a).

Ectopic expression of IL-32γ or treatment with cytokines reduced the expression of HNF1α and HNF4α at both the mRNA and protein levels, while they failed to significantly affect C/EBPα or HNF3β expression in the presence of HBV (Fig. 4b–d). Furthermore, confocal microscopy analysis demonstrated that only the cells expressing a high amount of IL-32γ showed the decreased levels of HNF1α and HNF4α expression (Fig. 4e, f).

Because the binding of HNF1α and HNF4α to HBV enhancers is essential for promoting viral transcription and replication[25], we tested whether the decrease in the levels of HNF1α and HNF4α induced by IL-32γ reduces their binding to cognate enhancer regions. We performed a ChIP assay using antibodies against HNF1α and HNF4α. Interestingly, our data revealed that HNF4α binding to the EnhI region was markedly reduced by IL-32γ, while its binding to EnhII/Cp was not affected (Fig. 4g). On the contrary, HNF1α binding to the EnhI region was not affected by IL-32γ, while its binding to EnhII/Cp was markedly reduced (Fig. 4h). To confirm that the binding of HNF4α to the EnhI region is altered in the presence of IL-32γ, we performed an EMSA. Similar to the results of the ChIP assay, we found that IL-32γ strongly reduced the binding of HNF4α to the EnhI region (Fig. 4i). The formation of the DNA–HNF4α complex was confirmed by western blotting. Taken together, our data suggest that IL-32γ decreases the expression of both HNF1α and HNF4α and consequently reduces their binding to viral enhancers.

**IL-32γ downregulates HNF4α through ERK1/2 pathways**. To elucidate the molecular mechanism by which IL-32γ downregulates the expression of HNF1α and HNF4α, we examined its effects on the signaling pathways related to HNF expression. As the expression of liver-enriched transcription factors is regulated by the MAPK signaling pathway[26,36], we investigated whether this pathway is involved in IL-32γ-mediated downregulation of HNF1α and HNF4α. Both IL-32γ overexpression and cytokine treatment strongly activated extracellular signal-regulated kinase (ERK)1/2 signaling but did not affect the phosphorylation of Jun N-terminal kinase (JNK) or p38 (Fig. 5a). Activation of ERK1/2 by IL-32γ correlated with the decreased expression of HNF1α and HNF4α. Furthermore, treatment with U0126, an inhibitor of the ERK1/2 pathway, completely restored the HNF4α expression and partially restored HNF1α expression (Fig. 5b).

Next, we determined the downregulation of which HNF is dominant in IL-32γ-mediated suppression of HBV. Supplementation with HNF4α almost completely restored viral replication (Fig. 5c), whereas supplementation with HNF1α only partially restored it (Fig. 5d). We further examined whether activation of ERK1/2 is involved in IL-32γ-mediated inhibition of HBV replication. Chemical inhibition of ERK1/2 activation markedly blunted the IL-32γ-mediated inhibition of viral replication (Fig. 5e), indicating that IL-32γ suppresses HBV replication through activation of the ERK1/2 pathway.

Taken together, our data demonstrate that IL-32γ downregulates the expression of HNF1α and HNF4α via the MAPK/ERK pathway, and that the reduced expression of HNF4α is the

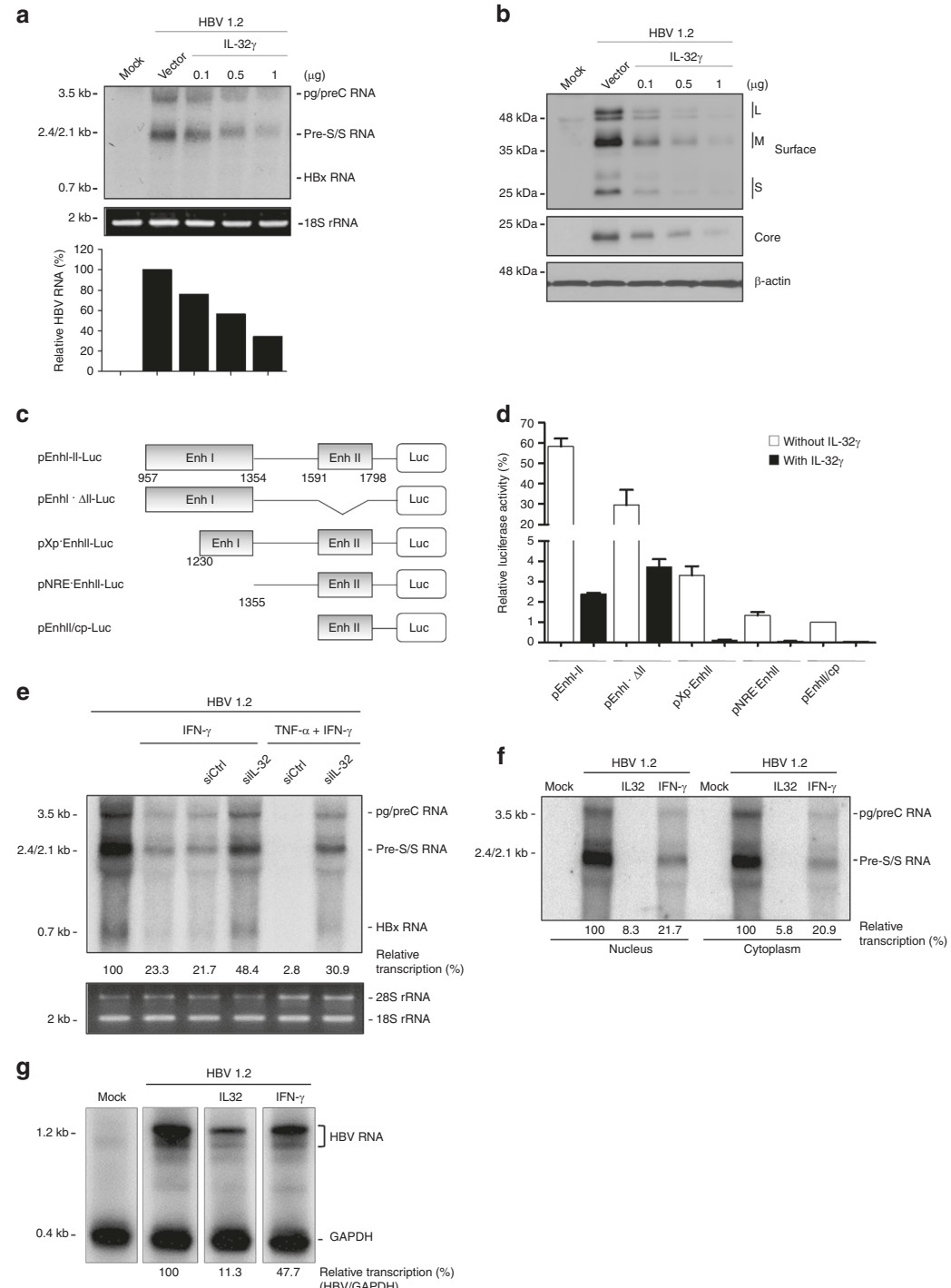

**Fig. 3** IL-32γ suppresses HBV transcription by downregulating viral enhancer/core promoter activities. **a**, **b** At 72 h post-co-transfection of Huh7 cells with HBV 1.2 and IL-32γ vectors, the levels of HBV RNAs and proteins were determined by Northern and western blotting, respectively. The 18S RNA was used as a loading control. **c** Cartoon of various HBV enhancer reporter mutants used in this study. **d** Effect of IL-32γ on each enhancer reporter mutant in Huh7 cells. Relative luciferase activity of each enhancer clone was determined at 48 h post-co-transfection with either empty or IL-32γ vector. **e** HBV 1.2 and siRNAs (20 nM) were co-transfected into Huh7 cells. Next day, the cells were treated with cytokines (TNF-α and IFN-γ) in fresh medium. At 72 h post-transfection, the cells were harvested and the levels of HBV RNAs were determined by Northern blot analysis. **f** Huh7 cells were co-transfected with HBV 1.2 and empty or IL-32γ vector. Next day, the cells were treated with IFN-γ. At 72 h post-transfection, the cells were harvested and nuclear and cytoplasmic fractions were isolated. Total RNA was extracted and subjected to Northern blot analysis. **g** Analysis of HBV transcription rate by nuclear run-on assay using the nuclear fractions shown in **f**. The level of HBV transcription was normalized to that of GAPDH RNA. Data (**d**) were obtained from three independent experiments (mean ± S.D.)

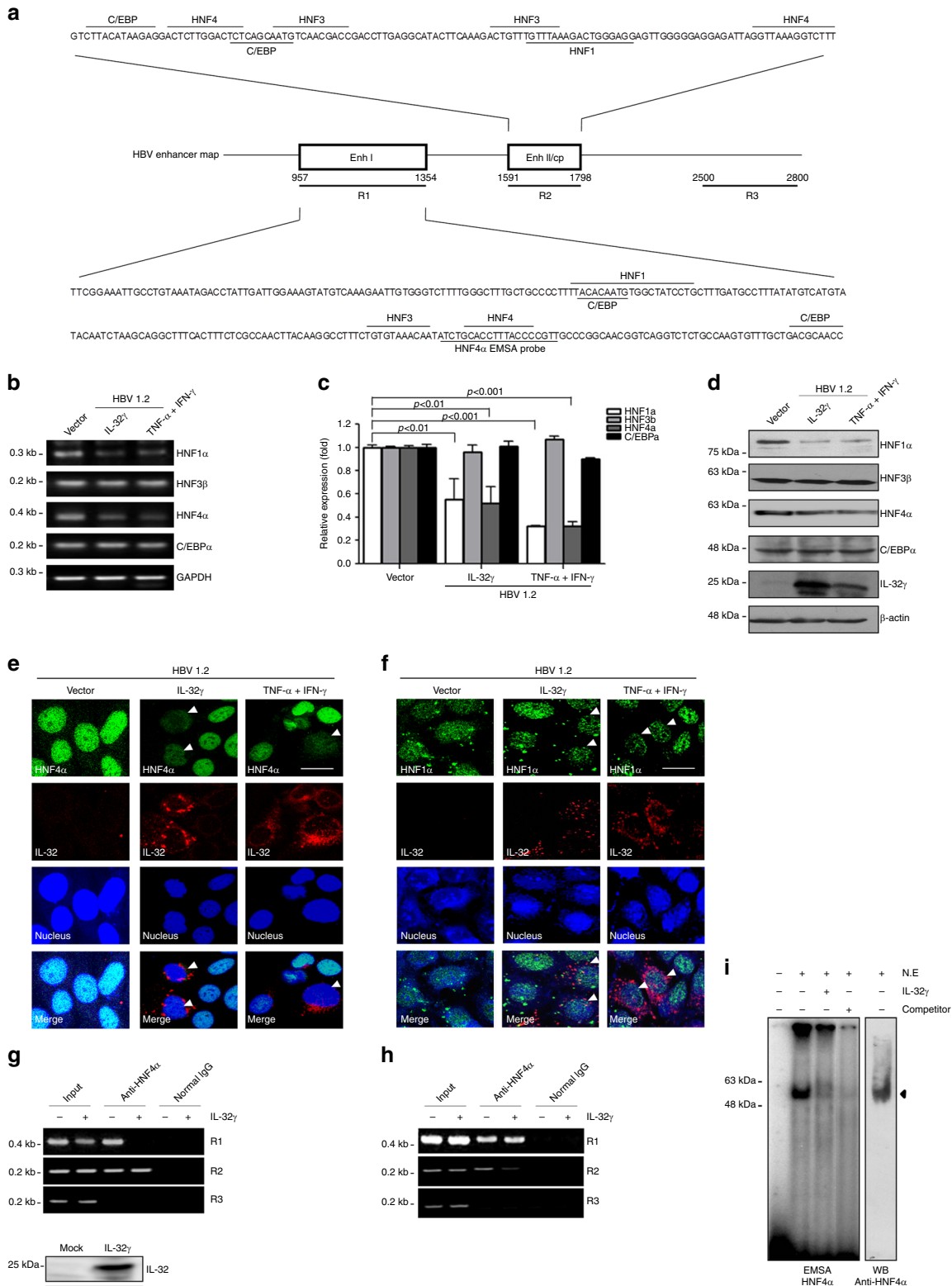

**Fig. 4** IL-32γ downregulates HNF1α and HNF4α and reduces their binding to enhancers. **a** Map of liver-enriched transcription factor binding to HBV enhancers. **b**–**d** Huh7 cells were co-transfected with the HBV 1.2 and IL-32γ vectors or were treated with cytokines after HBV 1.2 transfection. The levels of transcription factors were determined by semi-quantitative RT-PCR (**b**), real-time PCR (**c**), or western blot analysis (**d**. **e**, **f**) Effect of IL-32γ and cytokines on the expression of HNF1α and HNF4α was analyzed by confocal fluorescence microscopy. At 24 h after transfection or treatment, immunofluorescence assay was performed using indicated antibodies (magnification, ×400; scale bar, 50 μm.). **g**, **h** Chromatin immunoprecipitation (ChIP) assay. Control or IL-32γ vector was transfected into Huh7 cells and ChIP assay was performed using anti-HNF4α (**g**) or anti-HNF1α (**h**) antibody. The level of the IL-32γ protein was determined by western blotting. The regions for R1–R3 were shown in above diagram (**a**). **i** Electrophoretic mobility shift assay (EMSA). An aliquot of 2 μg nuclear extracts was used. A cold competitor (50-fold) was used as a negative control. The protein complex was confirmed by western blotting using anti-HNF4α antibody. Data (**c**) was obtained from three independent experiments (mean ± S.D.). $p < 0.001$, $p < 0.01$ by Student's $t$-test

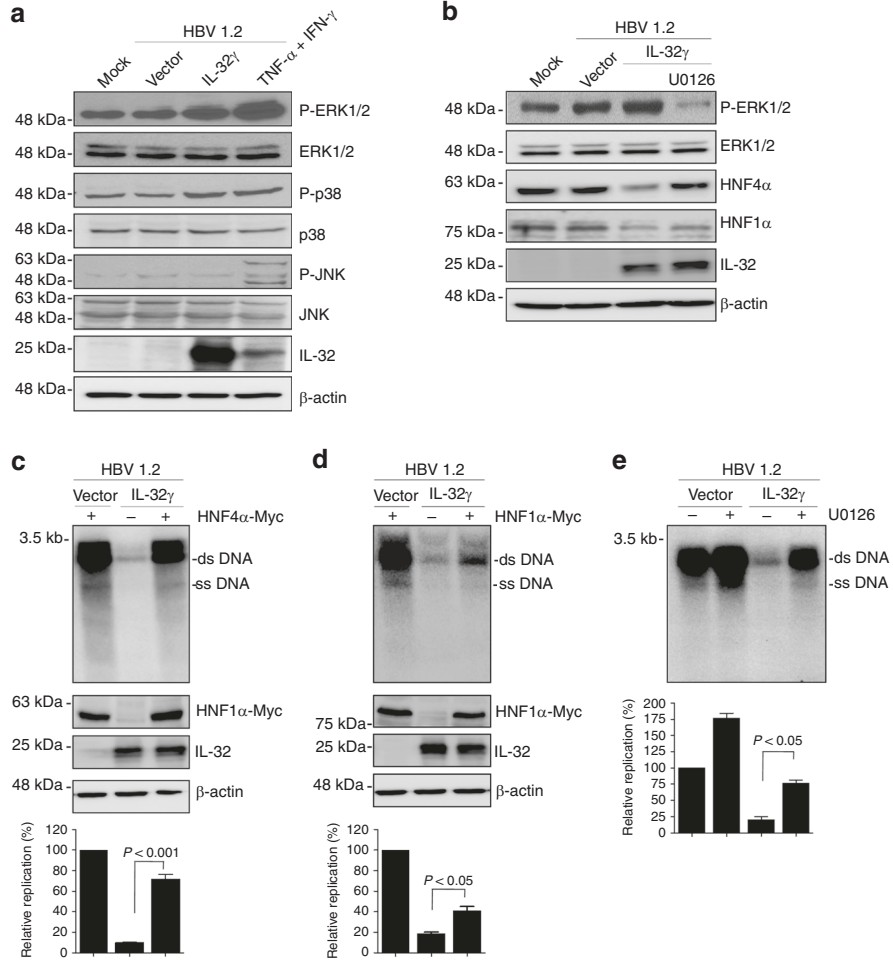

**Fig. 5** Involvement of ERK1/2 signaling in IL-32γ-mediated inhibition of HBV replication. **a** Effect of IL-32γ and cytokines on MAPK signaling in Huh7 cells. TNF-α and IFN-γ were added for 30 min before harvest. Activation of MAPK signaling pathways was determined by western blotting with the indicated antibodies. **b** Effect of ERK1/2 activation inhibitor (10 μM U0126) on the expression of HNFs. **c–e** Effect of HNF4α (**c**), HNF1α (**d**), and U0126 (**e**) on IL-32γ-mediated inhibition of HBV replication. Viral replication and protein expression were determined by Southern and western blot analyses, respectively. U0126 was added for 48 h at a final concentration of 10 μM. Data (**c–e**) were obtained from three independent experiments (mean ± S.D.). $p < 0.001$, $p < 0.05$ by Student's $t$-test

main cause of IL-32γ-mediated inhibition of HBV transcription and replication.

**IL-32γ suppresses HBV in a mouse model**. To investigate whether IL-32γ suppresses HBV in vivo, we established an HBV mouse model by using a hydrodynamic injection method[37,38]. After hydrodynamic injection of a plasmid carrying replication-competent HBV (HBV 1.2) and a plasmid for IL-32γ expression into the tail vein, mouse liver tissues and sera were analyzed. Expression of IL-32γ effectively suppressed both HBV replication in the liver and HBsAg secretion into serum (Fig. 6a, b). Immunohistochemical analysis showed that the expression of the HBV core protein was also significantly decreased in the liver (Fig. 6c).

As we showed that IL-32-mediated inhibition of HBV replication is an intracellular event, if there is no paracrine effect, the IL-32 and HBV plasmids transduced by hydrodynamic injection should be mostly co-localized in same hepatocytes. To check this, we transduced a mixture of RFP and GFP plasmids (1:1) into mouse liver and analyzed whether the two proteins are co-expressed in the same hepatocytes (Fig. 6d). Indeed, RFP and GFP were co-expressed in most of the transduced hepatocytes. Our observation is consistent with the previous reports that the

two plasmids delivered by hydrodynamic injection were co-localized in 91% of hepatocytes examined[39,40].

In addition, when a mixture of HBV and β-gal plasmids (with or without IL-32γ) was hydrodynamically injected into mouse liver, immunofluorescence analysis showed that the HBV core protein and β-gal were co-localized in most of the transduced hepatocytes (Fig. 6c). Although the expression of the core protein was suppressed by IL-32, it seems evident that two proteins were co-localized in same hepatocytes. These results indicate that IL-32γ strongly suppresses HBV in vivo and the IL-32-mediated inhibition of HBV in mouse liver is due to the co-expression of IL-32 and HBV in the same hepatocytes, not to the paracrine effect of IL-32.

**Cytokine-induced IL-32γ suppresses HBV in PHHs**. Finally, to verify the physiological relevance of our findings, we validated the above results using PHHs infected with HBV. To determine the optimal HBV infection levels, PHHs were incubated with HBV inoculum (100–2000 HBV Geq per cell) and the expression levels of the surface (HBsAg) and core (HBcAg) antigens were analyzed at 9 days post-infection (dpi). As shown in Supplementary Figure 10, ~80% of PHHs were infected at 1000 HBV Geq per cell; thus, this condition was used for further studies.

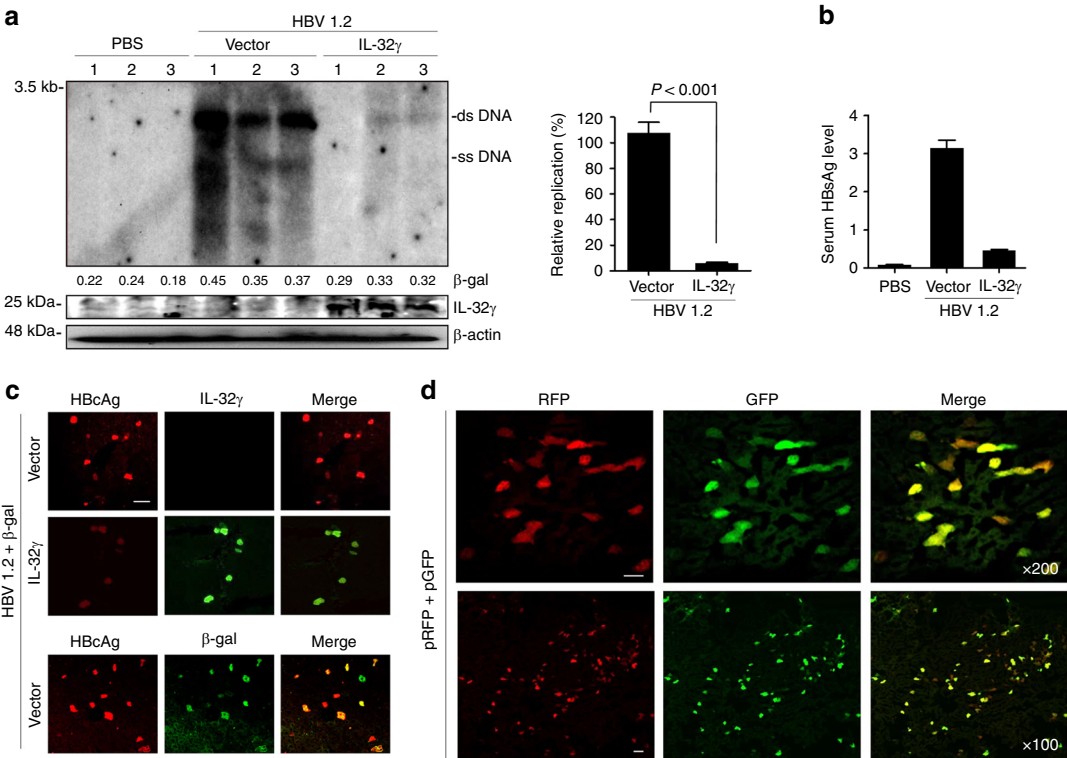

**Fig. 6** IL-32γ suppresses HBV in a mouse model. **a** Six-week-old male mice (3 mice per group) were killed 4 days after hydrodynamic injection of the indicated plasmids: control vector, 20 μg; HBV 1.2, 20 μg; β-gal, 5 μg. HBV replication was analyzed by Southern blotting using 50 mg of liver tissues. The level of IL-32γ and β-actin were analyzed by western blotting. **b** The level of secreted HBsAg in serum was measured by ELISA. **c** Immunofluorescence analysis of the HBV core, β-gal, and IL-32γ proteins in mouse liver tissues prepared as in **a**. β-gal protein staining was used as a control for hydrodynamic injection. Magnification, ×200; scale bar, 50 μm. **d** Immunofluorescence analysis of the co-expression of RFP and GFP proteins in the same hepatocytes. Each plasmid (pEF1α-RFP and pEF1α-GFP; 10 μg) was hydrodynamically injected into mouse liver. Sections were prepared with a tissue cryotome. Green and red fluorescence signals were observed with an inverted fluorescence microscope at indicated magnification (scale bar, 50 μm). Data (**a**, **b**) were obtained from three independent experiments (mean ± S.D.). $p < 0.001$ by Student's $t$-test

To assess the involvement of IL-32γ in TNF-α-mediated and IFN-γ-mediated suppression of HBV, we designed the experimental scheme described in Fig. 7a. At 4 dpi, the detection of similar levels of secreted HBeAg demonstrated that the level of HBV infection in each well was similar (Fig. 7a, bottom). At 6 dpi, IL-32 was knocked down by infection with shIL-32 lentivirus and the cells were treated with cytokines as described in Fig. 7a. First, we compared the induction level of IL-32γ by TNF-α or IFN-γ between Huh7 cells and PHHs. Importantly, PHHs produced a much higher level of IL-32γ (~55-fold) than did Huh7 cells both under unstimulated and cytokine-stimulated conditions (Supplementary Fig. 11). Suppression of HBV by cytokines in PHHs was confirmed by reduced HBsAg secretion (Supplementary Fig. 11).

To efficiently knockdown the expression of IL-32γ in PHHs, we infected Huh7 cells with four different shIL-32 lentiviruses and selected two lentiviral clones, sh32(2) (less effective) and sh32(4) (most effective) for further study (Supplementary Fig. 12). The effect of sh32(4) lentivirus was confirmed by the rescue of the cytokine-mediated suppression of HBV replication in HepG2 cells (Supplementary Fig. 13).

The knockdown of cytokine-induced IL-32γ completely restored the expression levels of HNF1α and HNF4α in PHHs (Fig. 7b). Using the same batch of cell lysates, we quantified HBV replication by real-time PCR (Fig. 7c). Importantly, IL-32γ knockdown remarkably rescued the TNF-α-mediated and IFN-γ-mediated inhibition of HBV replication and HBeAg expression (Fig. 7c, d). Under these experimental conditions, the overall levels of HNF1α and HNF4α were correlated with the levels of

HBV replication and secreted HBeAg. These results demonstrate that IL-32γ mediates the anti-HBV activity of TNF-α and IFN-γ in PHHs. Collectively, our data indicate that IL-32γ has a significant role in cytokine-mediated suppression of HBV during the natural course of HBV infection.

## Discussion

In the present study, we demonstrated a novel function of IL-32γ in hepatocytes, where it acts as a non-cytokine-like molecule: it mediates the anti-HBV activity of TNF-α and IFN-γ intracellularly. Based on our results, we propose a model of cytokine-mediated suppression of HBV (Fig. 7e). When hepatocytes are infected with HBV, the nearby hepatic immune cells secrete TNF-α and IFN-γ to suppress HBV in a non-cytopathic manner[4]. These cytokines synergistically induce the expression of IL-32γ, which in turn downregulates the expression of HNF1α and HNF4α, the transcription factors essential for HBV transcription, through the ERK1/2 pathway. Our findings may provide a mechanism of cytokine-mediated non-cytopathic viral clearance.

Generally, cytokines are crucial in host defense against pathogens and are released from immune cells and affect the same or nearby cells through autocrine or paracrine pathways. IL-32 was also identified as a cytokine secreted from NK cells[12,13]. Unexpectedly, we found that IL-32γ was not secreted at all not only in liver cell lines but also in PHHs and differentiated HepaRG cells when it was expressed by either transfection or induction by cytokines (Figs. 1f, g, and 2b), and strongly

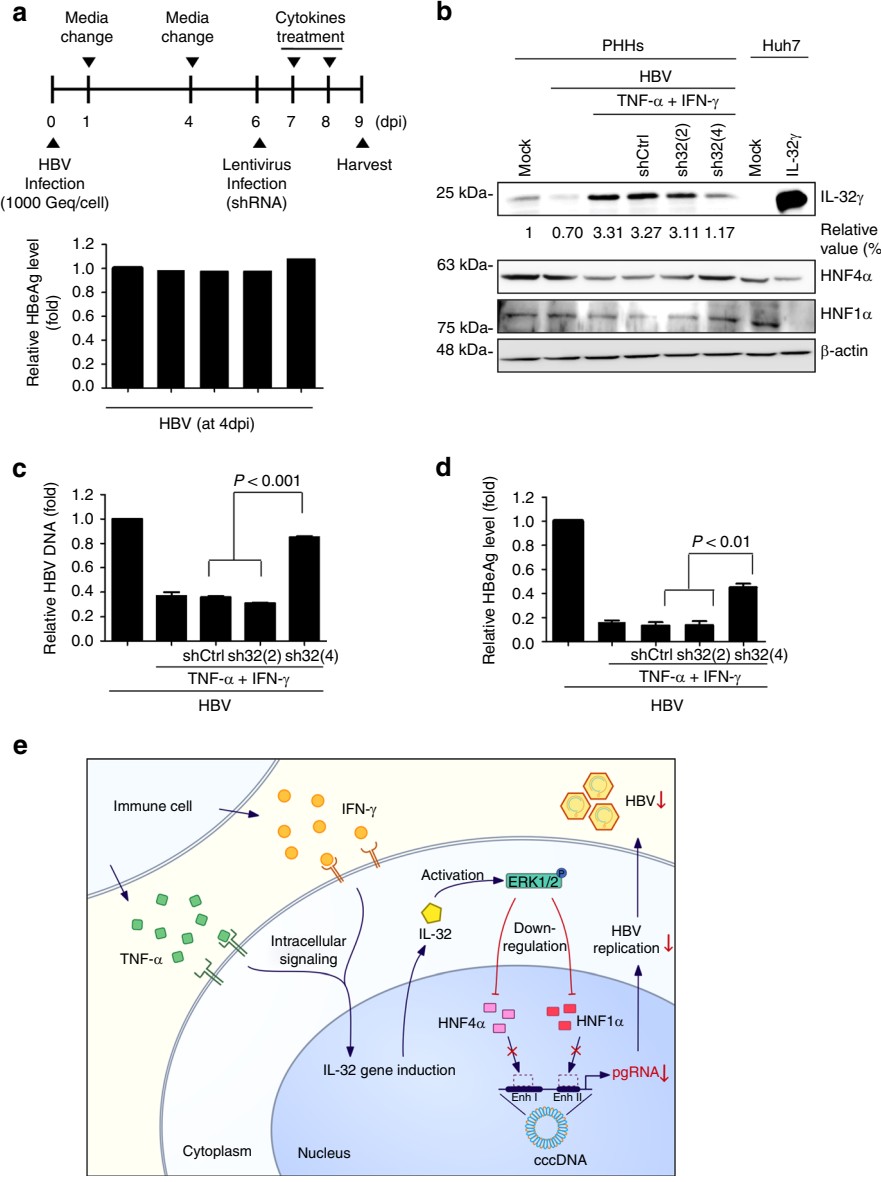

**Fig. 7** IL-32γ is involved in cytokine-mediated suppression of HBV in PHHs. **a** The experimental scheme. PHHs were infected with 1000 genome equivalents HBV per cell. At 4 days post-infection (dpi), the HBeAg level was determined by ELISA. **b** Effect of IL-32 knockdown on cytokine-induced downregulation of HNF1α and HNF4α. PHHs were infected with HBV and shIL-32 lentivirus as described in **a**. Cytokines were added for 2 days before harvest. **c**, **d** Effect of IL-32 knockdown on cytokine-induced inhibition of HBV replication and HBeAg secretion. Aliquots of cell lysates were used for real-time PCR and HBeAg ELISA. **e** A hypothetical model of IL-32γ-mediated suppression of HBV through downregulation of HNF1α and HNF4α expression. When hepatocytes are infected with HBV, immune cells secrete TNF-α and IFN-γ, which induce the expression of IL-32γ. IL-32γ activates ERK1/2, which in turn downregulates the expression of HNF1α and HNF4α. Finally, the binding of HNFs to the viral enhancers/core promoter is reduced, which consequently inhibits HBV transcription and replication. Data (**c**, **d**) were obtained from three independent experiments (mean ± S.D.). $p < 0.001$ by Student's $t$-test

suppressed HBV replication. These data demonstrate that IL-32γ functions through a cytoplasmic event, not a paracrine or autocrine pathway, suggesting that IL-32γ functions as a non-cytokine-like molecule in HBV suppression, at least in hepatocytes.

Enormous efforts have been dedicated to the identification of the receptor for IL-32; however, to date, the identity of this receptor is still unclear. In our study, extracellular treatment with active rhIL-32γ had no effect on HBV, whereas ectopic or cytokine-induced expression of IL-32γ strongly suppressed HBV replication (Figs. 1 and 2). As the cytokine-mediated anti-HBV activity of IL-32 occurs entirely in the cytoplasm, our finding

suggests that the IL-32 receptor is not necessary to mediate cytokine-induced viral suppression in hepatocytes.

Six alternative splice variants of IL-32 mRNA have been identified; among them, IL-32γ is the most active form in terms of inducing proinflammatory cytokines[15]. IL-32α, β, and γ share common exons. The difference in the amino acid sequence between them is in the short N-terminal and C-terminal regions (Fig. 2c). In this study, intracellular expression of any of the three representative IL-32 isoforms (IL-32α, IL-32β, and IL-32γ) suppressed the replication of HBV, with the order of the antiviral activity: IL-32γ > IL-32β > IL-32α (Fig. 2d). These data suggest that the additional C-terminal and N-terminal domains in IL-32γ

are responsible for strong anti-HBV activity. However, ectopically expressed IL-32γ did not induce the representative proinflammatory cytokines (TNF-α, TGF-β, IL-1β, IL-6, IFN-α, IFN-β, and IFN-γ) in hepatocytes (Supplementary Fig. 6b). These results suggest that IL-32γ activity in cytokine induction is totally different from that in the antiviral response. Nevertheless, it is notable that IL-32γ is the most active form both in the induction of proinflammatory cytokines and antiviral activity. Deciphering why IL-32γ is the most active form may provide a clue to its behavior as a cytokine and a non-cytokine-like molecule.

Several cytokines inhibit HBV replication through diverse mechanisms;[4,5] Among them, TNF-α and IFN-γ are secreted mainly by T cells and suppress HBV in a non-cytolytic manner[1,3,8]. TNF-α inhibits HBV RNA production and decreases capsid stability[1–3,6]. We also showed that p22-FLIP mediates the suppression of HBV by TNF-α[10]. IFN-γ reduces the pregenomic RNA-containing capsids[5]. Hepatocystin augments the IFN-γ-mediated suppression of HBV[11]. Several cytokine-induced antiviral proteins such as ISGs, OAS, MX, and CIAP2 are involved in TNF-α-induced or IFN-γ-induced inhibition of HBV;[41–43] however, these antiviral proteins seemed not to be related to the anti-HBV activity of IL-32γ (Supplementary Fig. 6a). Importantly, TNF-α and IFN-γ have been reported to reduce the levels of HBV cccDNA in hepatocytes[9]. In this study, we showed that IL-32γ is strongly induced by both TNF-α and IFN-γ in hepatocytes and mediates the inhibition of HBV by these cytokines by downregulating HNFs (Figs. 1 and 7).

It is well established that cytokines inhibit HBV at the posttranscriptional level in a mouse model[30]. The stability of HBV RNA is reduced by cytokine-induced depletion of the RNA stabilizing La protein[31–33] or destabilization of HBV RNA-containing capsids via proteasome[44,45] or kinase activity[46]. Therefore, we carefully examined whether there is any difference between mouse and human hepatocytes in terms of the cytokine effect on HBV suppression. As shown in Supplementary Figures 7, 8, and 9, cytokines appear to suppress HBV RNAs through a La-dependent pathway in mouse hepatocytes but through a La-independent pathway in human hepatocytes. There was no restoration of IL-32-mediated downregulation of HBV RNA or antigens when we treated Huh7 cells with the same inhibitors of JAK (AG490) and proteasome (MG132) that were used in previous studies (Supplementary Fig. 14). These results demonstrate that the antiviral effect of IL-32 is independent of proteasome or kinase activity at least in human hepatocytes. Altogether, our results suggest that cytokines suppress HBV at the posttranscriptional level in mouse hepatocytes, mainly through a La-dependent pathway; however, they suppress HBV at the transcriptional level in human hepatocytes through an IL-32-dependent pathway.

Importantly, intensive searching with sequences of all IL-32 isoforms did not reveal a homologous mouse gene[12]; therefore, studying the role for endogenous IL-32 in mice is not possible. To confirm the absence of IL-32 in mice, we isolated PMHs and confirmed that IL-32 is not detectable in these cells upon cytokine induction (Supplementary Fig. 7d). The difference in the mechanism involved in cytokine-mediated suppression of HBV between human and mouse is probably due to the fact that the mouse does not have any IL-32 homologs and does not support HBV infection.

During acute HBV infection, the non-cytopathic viral clearance of HBV is mediated by TNF-α and IFN-γ secreted mainly by T cells and is achieved through the inhibition of viral transcription and replication[1] or reduction of HBV cccDNA through the induction of APOBEC3[9]. We found no morphological or biochemical evidence that either treatment with TNF-α and IFN-γ or overexpression of IL-32γ induced cell damage or death (Figs. 1

and 2). Moreover, the mouse liver tissues overexpressing IL-32γ, which resulted in HBV suppression, were intact, without any histological signs of tissue damage (Fig. 6c), suggesting that HBV was suppressed through a non-cytolytic pathway. These findings may provide a mechanism of non-cytopathic viral clearance by cytokines or the induction of IL-32γ may contribute to the non-cytopathic clearance of HBV.

It is worth noting that the basal level of IL-32γ is much higher in PHHs (~55-fold) than in Huh7 cell (Supplementary Fig. 11). Moreover, IL-32γ is highly induced by TNF-α or IFN-γ treatment in PHHs. These observations support the possibility that IL-32γ has an important role during acute HBV infection and is involved in cytokine-mediated suppression of HBV during the natural course of HBV infection.

TNF-α is induced from human hepatocytes in chronic viral hepatitis[47] and the levels of TNF-α and IFN-γ in serum of acute hepatitis B patients are higher than in healthy controls[9]. Together with these reports, the observation that the expression level of IL-32γ in chronic HBV-infected liver was increased[48] may emphasize the clinical relevance of our finding that IL-32 is induced by TNF-α and IFN-γ in infected hepatocytes.

Single nucleotide polymorphisms in the promoters of TNF-α and IFN-γ genes affect the expression of these cytokines and are associated with HBV clearance[49,50]. As IL-32γ was induced by treatment with TNF-α and IFN-γ (Figs. 1 and 7), promoter polymorphisms in TNF-α and IFN-γ may correlate with the expression level of intrahepatic IL-32γ, which may contribute to HBV clearance.

As the TNF-α-induced activation of NF-κB signaling[51] and the nitric oxide pathway inhibit HBV replication[52], we tested whether TNF-α-induced IL-32γ inhibits HBV replication through these pathways. Neither activation of NF-κB nor production of nitric oxide appeared to be involved in IL-32γ-mediated inhibition of HBV (Supplementary Fig. 15). Nevertheless, our data showed that IL-32γ activates ERK1/2 signaling and controls HNF expression. These data are consistent with the previous studies that HNFs are regulated by the MAPK signaling pathway[26,36].

Previously, Li et al.[23] showed that IL-32 has no antiviral effect in HepG2.2.15, Huh7, L02, and Hep3B cells. However, our data clearly show that IL-32 inhibits HBV replication in HepG2, Huh7, L02, and Hep3B cells (Supplementary Fig. 3). Interestingly, similar to their study, IL-32 showed no antiviral effect in HepG2.2.15 and HepAD38 cells where HBV was stably expressed. However, HBV enhancer activity was commonly reduced in both cell lines when they were transfected with an enhancer reporter (Supplementary Fig. 16). A similar difference in the effect of IFN on HBV transcription was observed between HepG2 and HepDE19 stable cell lines[29]. The expression of CMV promoter-driven GFP which was transfected as a control was not affected by IL-32 (Fig. 2a and Supplementary Fig. 3a). These results suggest that IL-32 specifically suppresses viral enhancers/promoters when the HBV genome is present in an episomal form, reminiscent of HBV cccDNA.

Although IL-32γ downregulated both HNF1α and HNF4α, supplementation with HNF1α had only a very small effect on HBV replication, whereas HNF4α almost completely restored it (Fig. 5c, d). These results suggest that the reduced expression of HNF4α has a major role in IL-32γ-mediated inhibition of HBV replication. This is probably because HNF4α is the master transcription factor in HBV replication[24,25] and partly because HNF4α is necessary for HNF1α production[53]. Interestingly, although the binding sites for HNF1α and HNF4α are present both in the EnhI and EnhII regions, IL-32γ specifically reduced the HNF4α binding to EnhI and HNF1α binding to EnhII/Cp (Fig. 4g, h). As the promoter activity of EnhI is much higher than that of EnhII/Cp (~30 fold) (Fig. 3d) and HNF4α is the master

regulator of HBV replication, the specific reduction in HNF4α binding to EnhI by IL-32γ (Fig. 4g) seems to be the optimal strategy for the host to maximize the anti-HBV activity of IL-32γ. How IL-32γ alters the binding of HNFs to specific enhancers needs further investigation.

IL-32γ does not appear to regulate the common antiviral genes or transcription factors, which have a critical role in replication of several viruses. IL-32γ suppresses the replication of VSV[16,17], HIV-1[18,19], and influenza virus[20–22] through an extracellular pathway. Although IL-32 is known to suppress influenza virus in lung cells[20–22], it showed no effect on its replication in hepatocytes (Fig. 2f and Supplementary Fig. 5). It activates the expression of several cytokines such as IL-6, TNF-α, and IFN-γ in immune cells and inhibits HIV-1[18]. However, these cytokines were not induced by IL-32γ in hepatocytes (Supplementary Fig. 6b). Although IL-32 is involved in hepatitis C virus (HCV)-related liver inflammation and fibrosis, it shows no inhibitory effect on the replication of HCV, another hepatotropic virus[54]. Taken together, the available data indicate that the antiviral activity of IL-32 is cell-specific and target virus-specific.

In summary, in this study we showed that intracellular IL-32γ mediates the anti-HBV activity of TNF-α and IFN-γ by reducing the expression of HNFs. Thus, IL-32γ directly suppresses HBV at the transcriptional level through a non-cytokine-like function in hepatocytes. Our finding may provide a mechanism of non-cytopathic viral clearance and could be useful for developing new therapeutic options for the control of hepatitis B.

## Methods

**Cell culture and transfection**. Human hepatoma cell lines (Huh7, Korean Cell Line Bank, KCLB, 60104; HepG2, American Type Culture Collection, ATCC, HB-8065; Hep3B, American Type Culture Collection, ATCC, HB-8064) and normal hepatocyte cell LO2 (The Cell Bank of Type Culture Collection of Chinese Academy of Sciences, CBTCCCAS) were grown in Dulbecco's modified Eagle's medium (DMEM) (Gibco BRL, Grand Island, USA) supplemented with 10% (v/v) heat-inactivated fetal bovine serum (Gibco BRL), 1% penicillin, and 1% streptomycin (Gibco BRL) at 37 °C in 5% CO$_2$ incubator Transient transfection of cell lines was performed at 70–80% confluency using Lipofectamine 2000 reagent (Invitrogen, Carlsbad, CA, USA) according to the manufacturer's instructions. Differentiated HepaRG cells (Biopredic, Saint-Gregoire, France) were maintained according to the supplier's protocol.

**Isolation of primary hepatocytes and transfection**. Primary mouse hepatocytes (PMHs) were isolated from mouse liver by using a two-step collagenase perfusion method[55]. The liver specimens (~0.5 cm × 0.5 cm) were perfused through vein vessels on the cut surface of the specimen with cold perfusion buffer supplemented with collagenase (0.5 g/L) and calcium chloride (0.56 g/L). The cells were filtered through stainless steel meshes in two steps (grid size, 300 and 150 μm). The cells were washed twice with cold William's medium[10,28]. Human liver tissue specimens proved negative for HBV and HCV infection were obtained from therapeutic hepatectomies. Informed consent was obtained from patients before the procedure. PHHs were isolated with the approval from the Institutional Review Boards at St. Mary's Hospital (IRB No. UC14TIS10131) and Korea University Hospital (IRB No. ED10287). For transient transfection of PMHs and PHHs, JetPEI Hepatocyte reagent (Polyplus, Salt Lake City, NY, USA) was used according to the manufacturer's protocols.

**Plasmid construction**. The plasmids for HBV 1.2[56], NF-κB-luc[57], and HBV enhancer-luciferase reporters[10] were used in our previous reports. The plasmids for pCAGGS control, IL-32α, IL-32β[58], and IL-32γ[59] (GenBank no: BC009401) were also described previously. The expression plasmids for pHNF4α-Myc and pHNF1α-Myc were constructed by PCR amplification with a HepG2 cDNA library as a template and subcloned into the pCMV-Myc vector (Clontech, Mountain View, CA, USA).

**Cell viability assay**. Cell viability was determined by using a viability assay kit (Welgene, Seoul, Korea). After transfection or treatment of cells, the culture media were transferred to 96-well plates. The XTT and PMS reagents were added and incubated for 1 h. Cell viability was determined using a spectrophotometer as absorbance at 450 nm.

**RNA interference**. The IL-32 siRNA was synthesized by ST Pharm (Seoul, Korea) as the sense (5′-GGCUUAUUAUGAGGAGCAGTT-3′) and antisense oligonucleotides (5′-CUGCUCCUCAUAAUAAGCCTT-3′). Knockdown of IL-32 was confirmed by western blot analysis (Supplementary Fig. 1). The SSB/La siRNA was purchased from Santa Cruz Biotechnology (sc-40915). The siRNAs were transfected into cells by using Lipofectamine 2000 (Invitrogen). A set of Sigma Mission shRNA lentiviral constructs (TRCN0000174112, TRCN0000372658, TRCN0000059216, and TRCN0000372720) and control shRNA (pGL2 plasmid) were used for IL-32 knockdown in the presence of the packaging plasmids VSV-G and delta 8.2. All shRNA vectors for IL-32 were confirmed by sequencing. Vectors for shRNAs, delta 8.2, and VSV-G were co-transfected into 293T cells with Lipofectamine 2000. Culture supernatants were collected 24 h after transfection and used to infect Huh7, HepG2, and PHH cells in the presence of 6 μg/mL polybrene for 12 h at 37 °C.

**Western blot analysis**. Western blot analysis was performed as follows[60]. Cells were harvested 2 or 3 days after transfection with indicated plasmids, and lysed by RIPA buffer [20mM Tris/HCl, 1% NP-40, 0.5% protease inhibitor cocktail (Sigma, St. Louis, MO), 150mM NaCl, 2mM KCl, pH7.4]. After spin-down cell debris, the lysates were separated by SDS-PAGE and transferred to PVDF membrane. Primary antibodies against the following proteins and epitopes were used: HBV core protein (Dako, B0586, Hamburg, CA, USA, 1:2000), GFP (Sigma, G6795, 1:2000), HBsAg (Abcam, ab9193, Cambridge, UK, or Dako, 1:2000), HNF1α (Santa Cruz Biotechnology, sc-8986, 1:2000), HNF3β (Santa Cruz Biotechnology, sc-101060, 1:2000), CEBPα (Santa Cruz Biotechnology, sc-9314, 1:2000), HNF4α (Santa Cruz Biotechnology, sc-6556, 1:2000), Myc (Abcam, ab39688, 1:2000), lamin (Santa Cruz Biotechnology, sc-376248, 1:2000), NP (AbD Serotec, Raleigh, NC, USA, 1:2000), ICP27 (Abcam, 1:2000), tubulin (Santa Cruz Biotechnology, sc-8035, 1:2000), and actin (Sigma, A5316, 1:5000). Reagents used in the present study included recombinant human IL-32γ (rhIL-32γ) (YbdY, Seoul, Korea), anti-IL-32 antibody (YbdY, PAB101, 1:2000), anti-SSb/La antibody (Santa Cruz Biotechnology, Dallas, TX, USA, sc-80656, 1:2000), human TNF-α (YbdY), human IFN-γ (LG, Jeonbuk, Korea), mouse TNF-α (mTNF-α) (Peprotech, Rocky Hill, NJ, USA, 315-01A), mouse IFN-γ (mIFN-γ) (Peprotech, 315-05), U0126 (Cell Signaling, Boston, New York, USA), AG490 (Sigma, St. Louis, MO, USA), and MG132 (Merck Millipore, MA, USA). Primary antibodies were used at 1:2000 dilution. Goat anti-mouse immunoglobulin G conjugated with horseradish peroxide (secondary antibody; Santa Cruz Biotechnology) was used; signals were detected by using the ECL Plus reagent (ELPIS, Taejeon, Korea), except that HNF1α was detected with the Femto Substrate (Thermo, Rockford, IL, USA). The same membranes were stripped and re-blotted to detect the levels of other proteins. The original uncropped figures can be found in the supplementary information (Supplementary Fig. 17–38).

**Luciferase reporter analysis**. The luciferase reporter plasmids harboring deletion mutants of HBV enhancer regions were constructed and activities of these reporters were determined as follows[10]. Approximately 2 × 10$^5$ cells cultured on 12-well plate were transfected with a plasmid mixture containing 0.5 μg enhancer-luciferase (pEnhI.II, pEnhI.ΔII, pXp.EnhII, pNRE.EnhII, or pEnhII/cp), 0.5 μg IL-32γ, and 0.25 μg β-gal. Control vector (pCAGGS) was used to normalize the amount of transfected DNA. After 48 h post-transfection, cells were harvested and subjected to measure luciferase activity using the Steady Glo-Luciferase system (Promega, Madison, WI). Data were obtained from at least three independent experiments.

**ELISA**. The expression level of IL-32 was measured with an IL-32 ELISA kit (YbdY). A total of 5 × 10$^5$ cells were seeded per well of a 12-well plate. After 48 h post-treatment with cytokines or transfection, cells were harvested, and 1/10 of culture medium or lysate of each group was used for ELISA. The final data were converted to the total amount of IL-32 per well of a 12-well plate. The expression levels of HBeAg and HBsAg were measured using an HBeAg/HBsAg ELISA kit (Wantai Pharm Inc., Beijing, China) according to the manufacturer's instructions. Data were obtained from at least three independent experiments.

**Infection with HBV, HSV, and IAV**. HBV infection was carried out as follows[10,28,57]. PHHs or differentiated HepaRG cells were infected with 1000 HBV genome equivalents per cell (Geq/cell) in fresh medium containing 4% PEG and 2.5% DMSO. At 15 h post-infection, the medium was replaced with fresh medium and the cells were harvested 9 days post-infection.

HSV-1 K26GFP, a recombinant herpes simplex type 1 expressing GFP (HSV-1_GFP) was obtained from Dr. Prashant Desai (Johns Hopkins University)[61]. The virus titer was determined by plaque assays with Vero cells overlaid with 1% methylcellulose (Sigma) in normal growth medium. At 3 days after infection, the cells were fixed and stained with 2% crystal violet (Merck Millipore) in 20% ethanol.

The influenza A virus H1N1 strain (A/WSN/33) eight-plasmid reverse-genetics system was kindly provided by Dr. Ren Sun[62]. A/WSN/33 expressing GFP (A/WSN/33_GFP) was generated with this supplemented with an enhanced GFP plasmid as previously described[63]. The virus titer was determined by TCID50 assay with MDCK cells according to published procedures[64]. Briefly, monolayers of MDCK cells in 96-

well cell culture plates were inoculated with 100 μL of virus inoculum and infected for 2 h. Cells were washed with serum-free medium and incubated with influenza A virus growth medium containing 2 μg/mL TPCK-trypsin (Sigma). At 3 days post-infection, cytopathic effect (CPE) of each well was counted.

Huh7 cells were transfected with the IL-32γ plasmid. At 16 h post-transfection, the cells were infected with HSV-1_GFP or A/WSN/33_GFP virus at 1 multiplicity of infection (MOI) (adsorption time, 1 h). After infection, the medium was replaced with fresh medium, and cells were incubated for 24 h. Treatment with ganciclovir (GCV) (Sigma) or a mixture of IFN-α (1000 U/mL) and IFN-γ (100 U/mL) was used as a positive control. The cells were monitored for green fluorescence signals and images were taken with an inverted fluorescence microscope (Leica DM IL LED Fluo, Leica, Wetzlar, Germany) at ×100 magnification.

**Southern and Northern blot analyses**. The replication of HBV was detected by Southern blot analysis[65]. Cells were harvested by scraping at 3 days post-transfection and lysed with 100 μL HEPES buffer. Viral capsids were precipitated with PEG buffer and digested with SDS buffer containing Proteinase K at 37 °C for 3 h. Total viral DNA was separated on a 1% agarose gel at 100 V for 3 h and transferred to a nitrocellulose membrane (GE healthcare). To detect HBV DNA, the membrane was hybridized with a highly purified [32P]-labeled HBV probes. The level of HBV transcription was determined by Northern blot analysis as follows. Total RNA was extracted by using TRIzol reagent (Invitrogen) according to the manufacturer's protocol. Total RNA (20 μg) was separated on a 1% for-maldehyde agarose gel at 120 V for 3 h and transferred to a nitrocellulose mem-brane (GE healthcare) for 18 h. To detect HBV-specific RNAs, the membrane was hybridized with a highly pure [32P]-labeled HBV probes. 18S RNA was used as a loading control. Relative levels of HBV replication and transcription were calcu-lated using a phosphorimager.

**Nuclear run-on assay**. To determine the rate of HBV transcription, nuclear run-on assay was performed[31]. Nuclear extracts were prepared from ~1 × 10^7 cells[34] and mixed with an equal volume of 2× transcription reaction mix to final concentrations of 25 mM Tris (pH 8), 100 mM KCl, 7 mM MgCl_2, 12.5% glycerol, 0.5 mM each rATP, rGTP, rCTP and 0.25 mM UTP, 10 mM creatine phosphate, 1 mM DTT, 10 μg/mL PMSF, and 250 μCi [32P]-labeled UTP. Transcription was carried out for 20 min at 26 °C. RNase-free DNase was then added and the samples were incubated for 10 min at 30 °C, followed by proteinase K digestion for 30 min at 45 °C. After addition of 1/10 volume of 3 M NaOAc (pH 4), total RNA was extracted with 500 μL of phenol/chloroform and precipitated with ethanol at −80 °C. Labeled RNA was resuspended in STE buffer (10 mM Tris [pH 8], 1 mM EDTA, 140 mM NaCl), and free nucleotides were removed by gel filtration on a Sephadex G-50 column. The labeled RNA was hybridized to membranes that contained the PCR-amplified HBV probes.

**Real-time quantitative PCR**. Real-time quantitative PCR was performed as fol-lows[38]. Reverse transcription reactions were performed using 2 μg of total RNA and MMLV reverse transcriptase (Intron Biotechnology) in the final reaction mixture volume of 20 μL. The synthesized cDNA was amplified by PCR under the following conditions: denaturation at 94 °C for 5 min, followed by 40 cycles of 94 °C for 30 s and 72 °C for 1 min, and a final extension at 72 °C for 5 min The sequences of specific primers used for real-time PCR are listed in Supplementary Table. 1. Real-time quantitative PCR amplification was performed in an ABI PRISM 7500 sequence detection system using the SYBR Green PCR master mix (Applied Biosystems). The results were expressed as an n-fold difference relative to the calibrator (RQ = $2^{-\Delta\Delta Ct}$).

**Chromatin immunoprecipitation (ChIP) assay**. ChIP assay was performed as follows[10]. Briefly, sonicated chromatin was pre-cleared using protein A-agarose and incubated with anti-HNF1α, anti-HNF4α, or normal rabbit IgG as a negative control. Immunoprecipitated DNA fragments were amplified by PCR. The sequences of primers used for ChIP assay are listed in Supplementary Table. 1.

**Electrophoretic mobility shift assay (EMSA)**. EMSA was performed as follows[60]. Briefly, 2 μg nuclear extracts of Huh7 cells were incubated with the gamma [32P]-labeled dsDNA probe for HNF4α binding. A set of probe sequences located in the EnhI region was designed: 5′-TCTGCACCTTTACCCCGTTG-3′ and 5′-CAACGG GGTAAAGGTGCAGA-3′. After binding on ice, the DNA–protein complex was analyzed by electrophoresis at a low temperature in 6% native polyacrylamide gel. Gel was dried at 70 °C for 30 min. The unlabeled competitor probe for HNF4α binding was added for 10 min before the addition of [32P]-labeled probe.

**Confocal microscopy**. To evaluate the expression levels of IL-32γ, HNF4α, and HNF1α, confocal microscopy analysis was performed[60]. Cells (Huh7 and PHHs) grown on cover slides were incubated overnight at 4 °C with primary antibodies (1:300) containing 3% BSA in PBS. Huh7 cells were washed with PBS and incu-bated with secondary antibody conjugated with Alexa 488 or Alexa 568 for 1 h at room temperature. Cells were then washed 3 times with PBS at room temperature, and Toppro-3 (1:500) was added to stain nuclear DNA. The level of HBV infection

in PHHs was determined by counting the HBcAg and HBsAg-positive cells under a fluorescence microscope.

**Immunofluorescence microscopy of mouse tissues**. Liver tissues were fixed in 10% formalin and embedded in paraffin. Slides with 5-μm tissue sections were deparaffinized in xylene and rehydrated in ethanol. Antigen retrieval was per-formed in citrate buffer (pH 6.0) using microwave treatment. Primary antibodies for HBcAg (HBV core protein, Dako) (1:800), IL-32 (1:100), and/or β-gal were incubated for 1 h in a humidified chamber at room temperature, followed by detection with secondary antibody (1:50) at room temperature by staining with a TSA Plus fluorescence kit (Perkin Elmer, Waltham, MA, USA). The slide was then placed in citrate buffer (pH 6.0) and heated using a microwave oven. The images were acquired with a fluorescence microscope at ×100 magnification.

**Hydrodynamic injection and immunohistochemical analysis**. Six-week-old male mice (BALB/C) were hydrodynamically injected with plasmids (HBV 1.2, IL-32γ, and β-gal) in PBS[38]. For co-localization assay, 10 μg of each RFP and GFP plasmid was hydrodynamically injected into mouse liver. The volume equivalent to 10% of mouse body weight was injected via tail veins with high pressure within 4–6 s[37]. All mouse experiments were approved by the Animal Care Committee of the Konkuk University.

**IL-32γ activity assay**. Human THP-1 monocytic cell line and murine Raw 264.7 macrophage cell line were used. Cells were maintained in complete RPMI 1640 medium (Invitrogen) containing 10% fetal bovine serum (Invitrogen) supple-mented with 100 U/mL penicillin and 100 μg/mL streptomycin. THP-1 cells were washed with DPBS (Invitrogen) and seeded in 96-well tissue culture plates at 2 × 10^5 cells per well in 100 μL per well complete RPMI 1640 medium and then incubated with either 100 μL of culture medium (control) or 100 μL of medium containing 100 or 200 ng/mL of recombinant human IL-32γ (final concentrations: 50 and 100 ng/mL, respectively; YbdY) at 37 °C in a 5% CO_2 humidified atmo-sphere. After 18 h of treatment, supernatants were collected, centrifuged at 2000 rpm and stored at −20 °C until cytokine assay by ELISA. Raw 264.7 cells were cultured in 96-well tissue culture plates at 5 × 10^4 cells per well overnight and then attached cells were used for assay. Raw 264.7 cells were washed and incubated with stimuli as described above. The supernatants from THP-1 were evaluated for human IL-8 and those from Raw 264.7 for mouse TNF-α. The amounts of human IL-8 and murine TNF-α were determined by sandwich ELISA according to the manufacturer's instructions (R&D Systems, Minneapolis, MN, USA).

**NF-κB luciferase assay and treatment with a NOS inhibitor**. After co-transfection of Huh7 cells with pNF-κB-Luc (500 ng), control pcDNA3.1(+) (250 or 500 ng), and IL-32γ plasmid (250 or 500 ng), NF-κB-dependent luciferase activity was measured using a Steady Glo-Luciferase assay system (Promega, Madison, WI, USA) according to the manufacturer's instructions. The values were normalized to β-galactosidase activity in the same cells. Data were obtained from at least three independent experiments. After co-transfection of Huh7 cells with the HBV 1.2 and IL-32γ plasmids, the nitric oxide synthase (NOS) inhibitor L-NAME (Sigma) was added (to 100 or 500 μM) for 72 h. Southern blot analysis was per-formed to determine the HBV DNA levels.

**Statistical analysis**. Data were obtained from at least three independent experi-ments (mean ± S.D.). All statistical calculations ($p < 0.05$, $p < 0.01$, and $p < 0.001$) were obtained using Microsoft Excel 2010 software or the Student's t-test in GraphPad Prism 5.

**Data availability**. The data supporting the findings of this study are available within the article and its Supplementary Information files and from the corre-sponding author on reasonable request.

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

## Acknowledgements

This study was supported by the National Research Foundation of Korea (NRF) grants funded by the Korea government (NRF-2017R1A2B3006335 and NRF-2016R1A5A2012284), and by the Korea Health Technology R&D Project through the Korea Health Industry Development Institute (KHIDI), funded by the Ministry of Health & Welfare (No. HI17C0874).

## Author contributions

Study conception and design: D.H.K., E.-S.P. and K.-H.K. (Konkuk). Acquisition of data: D.H.K., E.-S.P., A.R.L., S.P., W.-C.C., Y.K.P., S.H.A., H.S.K., J.H.W., Y.N.H. and B.J. Analysis and interpretation of data: M.J.S., S.H.P. and S.-H.K. Material support: Y.K.P., S.H.A., H.S.K., A.R.L., S.P., J.H.W., Y.N.H., B.J., D.-S.K. and K.-H.K. (Catholic). Obtained funding: K.-H.K. (Konkuk) and E.-S.P. Drafted the manuscript: D.H.K., E.-S.P. and K.-H.K. (Konkuk).

## Additional information

**Competing interests:** The authors declare no competing interests.



