## [Peer Review File · Nature Communications]

Reviewers' comments:

Reviewer #1 (Remarks to the Author):

In this manuscript, Kim et al found that intracellular IL-32 γ was strongly induced by TNF- α and IFN- γ in hepatocytes and inhibited HBV replication and gene expression. Interestingly, the antiviral effect of IL-32 γ was restricted intracellularly by induction or overexpression, not by direct extracellular stimulation. Moreover, they identified that IL-32 γ activated ERK pathway to downregulate transcription factor HNF4A, thereby to inhibit HBV transcription. This study is innovative and interesting but there are several questions or suggestions need to be considered below:

1. As shown in Fig. 2B, overexpression of IL-32 γ by plasmid was able to produce IL-32 γ in supernatant. However, TNF- α and IFN- γ stimulation could only induce IL-32 γ in the cytoplasm. Please give an explanation for this phenomenon.
2. It is very interesting that IL-32 γ is expressed predominantly in cytoplasm. As shown in Fig.2D, other isoforms of IL-32 were also able to inhibit HBV replication. What is the difference between these isoforms? It's better to identify the functional elements in IL-32 gene which contribute to HBV inhibition and intracellular location.
3. Previously, Li et al found that IL-32 γ have no antiviral effect in HepG2.2.15 cells, as well as L02, Huh7 and Hep3B cells (J Biol Chem. 2013 Jul 19;288(29):20927-41.). Please discuss the different observation in same cell lines in the discussion part.
4. As suppl. Fig 8 showed that basal IL-32 expression in PHH was much higher (\sim 55 fold) than in hepatoma cell lines. Actually, the anti-HBV effect of IL-32 overexpression in mouse in vivo was described before (J Biol Chem. 2013 Jul 19;288(29):20927-41.). Here, I may suggest to ask the authors to knock down endogenous mouse IL-32 expression to examine whether HBV replication was increased or not.
5. Line72-74: "During the early Noncytopathic clearance". This sentence was not accurate and lack of references.
6. Which method was used for IL-32 γ detection in supernatant? The unit of IL-32 γ used in Fig 1F,G and other figures seems not correct.
7. The quality of HBV core protein western blot in Fig. 3B is bad and please repeat the experiment.
8. Line 319-320. "A literature search..... EnhII/Cp". This sentence was not accurate and lack of references. As reviewed by Quasdorf M. et al (J Viral Hepat. 2010 Aug;17(8):527-36), there are more TFs than these four genes.
9. line 385-386: "Suppression of HBV by cytokines....." The figure was missing.

Reviewer #2 (Remarks to the Author):

Kim et al. report here that interleukin (IL)-32 is an intracellular downstream mediator involved in the noncytopathic inhibition of HBV induced by IFN- γ and TNF- α . The Authors build on a growing body of literature that shows that IL-32 inhibits the replication of several viruses, including VSV, HIV and influenza. Kim et al. claim that the mechanism of action whereby IL-32 inhibits HBV occurs at the transcriptional level, and involves a decreased viral enhancer activity by ERK1/2-

dependent down-regulation of liver-enriched transcription factors. Although potentially of interest, there are a number of conceptual and technical issues that reduce my enthusiasm for the work. Specifically:

- 1) The claim that the IL-32-mediated inhibition of HBV occurs at the transcriptional level is at odds with previously published literature (the Authors are referred to Guidotti and Chisari, Annual Review of Pathology 2006 for a comprehensive review of those studies). Briefly, the noncytopathic inhibition of HBV replication by IFN-g was shown to occur at the post-transcriptional level, with minimal or no effect on HBV transcription. IFN-g was shown to prevent the assembly of replication-competent HBV RNA-containing capsids in the hepatocyte in a proteasome- and kinase-dependent manner. The viral nucleocapsids disappear from the cytoplasm of the hepatocytes, and the viral RNAs are destabilized by a SSB/La-dependent mechanism in the nucleus. How can the Authors reconcile such discrepancies? Have they tested a potential post-transcriptional activity of IL-32?
- 2) If the HBV-inhibiting activity of IL-32 is exclusively intracellular, how do the Authors explain the results reported in Fig. 6? The in vivo transduction efficiency of the hydrodynamic injection of a plasmid is known not to exceed 20% of the hepatocytes. Are we to assume that both HBV and IL-32 (encoded by two separate plasmids) randomly transduced the exact same hepatocytes?
- 3) In Fig. 2E-F, the Authors claim that IL-32 does not have any effect on the replication of other viruses, such as HSV-1 or influenza. To make such a claim, the Authors should select viruses and design experimental conditions whereby the same combination of TNF-a and IFN-g shown to inhibit HBV in Fig. 1 inhibits replication of said viruses as well.
- 4) In the Introduction, the Authors state: "During the early stage of HBV infection, HBV replication is suppressed by innate immunity through antiviral cytokines without damage to infected hepatocytes, a phenomenon dubbed noncytopathic clearance." The alleged role of innate immunity in viral clearance during natural HBV infection is unsubstantiated and contrasts our current understanding of the immune response against HBV.
- 5) The Authors are encouraged to provide comprehensive methodological details to allow a thorough understanding of the experiments. In many instances, I could not find such information in the Results, Methods or Figure legend section.

Minor:

- 1) Can the Authors provide a positive control for the anti-IL-32 blocking antibody (e.g. by showing that it effectively blocks IL-32 activity in the assay reported in Supplementary Fig. 2)?
- 2) Does IL-32 induce TNF-a and IFN-g?
- 3) Why do the mice injected with the IL-32 plasmids stain positive for b-gal?

Response to reviewer' comments

Reviewer #1: In this manuscript, Kim et al found that intracellular IL-32 γ was strongly induced by TNF- α and IFN- γ in hepatocytes and inhibited HBV replication and gene expression. Interestingly, the antiviral effect of IL-32 γ was restricted intracellularly by induction or overexpression, not by direct extracellular stimulation. Moreover, they identified that IL-32 γ activated ERK pathway to downregulate transcription factor HNF4A, thereby to inhibit HBV transcription. This study is innovative and interesting but there are several questions or suggestions need to be considered below:

1. As shown in Fig. 2B, overexpression of IL-32 γ by plasmid was able to produce IL-32 γ in supernatant. However, TNF- α and IFN- γ stimulation could only induce IL-32 γ in the cytoplasm. Please give an explanation for this phenomenon.

Response: Thank you for commenting. Considering that the level of overexpressed IL-32 γ is much higher than that of IL-32 induced by cytokines (Fig. 5a), we believe the secretion of small amounts of IL-32 into culture medium is a side effect caused by overexpression. Indeed, the ratio of IL-32 (intracellular vs. supernatant) is >100-fold in Huh7 cells (Fig 2b) and >25-fold in HepaRG and PHHs (Fig. 1g).

2. It is very interesting that IL-32 γ is expressed predominantly in cytoplasm. As shown in Fig.2D, other isoforms of IL-32 were also able to inhibit HBV replication. What is the difference between these isoforms? It's better to identify the functional elements in IL-32 gene which contribute to HBV inhibition and intracellular location.

Response: IL-32 α , β , and γ share common exons. The difference in the amino acid sequence between them is in the short N- and C- terminal regions (Fig. 2c). Compared to the smallest isoform, IL-32 α , IL-32 β harbors an additional domain in the C-terminal region, whereas IL-32 γ harbors additional domains in both the C- and N-terminal regions. Considering the order of antiviral activity (IL-32 γ > β > α), we think that the additional C- and N-terminal domains in IL-32 γ are responsible for strong anti-HBV activity. All isoforms of IL-32 tended to localize inside the cells, so we think that the common domains are responsible for intracellular localization at least in hepatocytes. To take into account the reviewer's

comment, we have added a schematic illustration of IL-32 isoforms in revised Fig. 2c. We discussed this in the Discussion section.

Figure 2c

3. Previously, Li et al found that IL-32 γ have no antiviral effect in HepG2.2.15 cells, as well as L02, Huh7 and Hep3B cells (J Biol Chem. 2013 Jul 19;288(29):20927-41.). Please discuss the different observation in same cell lines in the discussion part.

Response: We appreciate your helpful comment. To validate whether IL-32 γ inhibits HBV replication in several hepatocyte-derived cell lines, we performed Southern blot analysis using L02, Huh7, and Hep3B cells. Our results revealed that ectopic expression of IL-32 γ strongly inhibits HBV replication in all these cell lines (Supplementary Fig. 3).

Li et al. showed that IL-32 γ has no antiviral effect in HepG2.2.15 cells. We also tested the effect of IL-32 γ on HBV replication. We found that IL-32 γ has no antiviral effect in HepG2.2.15 and HepAD38 cell lines where HBV is stably expressed (Supplementary Fig. 16a and 16b). However, HBV enhancer activity was reduced in both cell lines when they were transfected with an enhancer reporter plasmid (Supplementary Fig. 16c and 16d). Considering that IL-32 suppresses the infected HBV in primary human hepatocytes (Fig. 7) and the transfected HBV plasmid in hepatocyte cell lines (Fig. 2 and Supplementary Fig. 3), our results suggest that IL-32 suppresses viral enhancers/promoters when the HBV genome is present in an episomal form, reminiscent of HBV cccDNA.

Interestingly, similar observations were also reported by Dr. Guo's group: IFN- γ suppressed HBV transcription in HepG2 cells transiently transfected with an HBV plasmid, but had no effect in cells stably transfected with HBV as shown below (Mao et al., 2011).

In addition, IL-32 showed no effect on the expression of CMV promoter-driven GFP which was transfected as a control (Fig. 2a and Supplementary Fig. 3a). These results further suggest that IL-32 specifically suppresses viral enhancers/promoters when the HBV genome

is present in an episomal form.

Following the reviewer's comment, we added the following sentences to the Discussion section: "Previously, Li et al. (2013) showed that IL-32 has no antiviral effect in HepG2.2.15, Huh7, L02, and Hep3B cell lines. However, our data clearly show that IL-32 inhibits HBV replication in HepG2, Huh7, L02, and Hep3B cells (Supplementary Fig. 3). Interestingly, similar to their study, IL-32 showed no antiviral effect in HepG2.2.15 and HepAD38 cells where HBV was stably expressed. However, HBV enhancer activity was reduced in both cell lines when cells were transfected with an enhancer reporter (Supplementary Fig. 17). A similar difference in the effect of IFN on HBV transcription was observed between HepG2 and HepDE19 stable cell lines (Mao et al., J Virol, 2011). The expression of CMV promoter-driven GFP was not affected by IL-32 (Fig. 2a and Supplementary Fig. 3a). These results suggest that IL-32 specifically suppresses viral enhancers/promoters when the HBV genome is present in an episomal form, reminiscent of HBV cccDNA."

(Mao et al, J virol, 2011)

Supplementary Fig. 3

Supplementary Fig. 16

4. As suppl. Fig 8 showed that basal IL-32 expression in PHH was much higher (~55 fold) than in hepatoma cell lines. Actually, the anti-HBV effect of IL-32 overexpression in mouse in vivo was described before (J Biol Chem. 2013 Jul 19;288(29):20927-41.). Here, I may suggest to ask the authors to knock down endogenous mouse IL-32 expression to examine whether HBV replication was increased or not.

Response: Thank you for your suggestion. According to Li et al. (2013), treatment of peripheral blood mononuclear cells (PBMCs) with recombinant IL-32 induced IFN- λ 1, which in turn inhibited HBV replication in hepatocytes indirectly. However, the direct antiviral action of IL-32 in hepatocytes has not been demonstrated. We think that the anti-HBV effect of IL-32 overexpression is due to the non-cytokine-like (direct) action of IL-32, not the induction of IFN- λ 1 (indirect action) in mice for the following reasons: 1) the IL-32 and HBV plasmids, which are delivered by hydrodynamic injection, is mainly transfected into hepatocytes, not the PBMCs where IFN- λ 1 is produced upon treatment with IL-32; 2) transfected hepatocytes do not secrete IL-32; therefore, it cannot induce IFN- λ 1 in PBMCs.

At first, we also tried to knock-down the expression of endogenous IL-32 in mice. However, intensive searching with sequences of all IL-32 isoforms did not reveal a homologous mouse gene (Kim et al., 2005); therefore, studying the role for endogenous IL-32 in mice is not possible. To confirm the absence of IL-32 in mice, we isolated primary mouse hepatocytes and confirmed that IL-32 is not detectable in these cells upon cytokine induction (Supplementary Fig. 7d). Therefore, instead of knock-down, we overexpressed IL-32 in primary mouse hepatocytes and observed that, like in human hepatocytes, IL-32 strongly suppresses HBV transcription and antigen production (Supplementary Figs. 7 and 9).

Supplementary Fig. 7

Supplementary Fig. 9

5. Line72-74: “During the early ... Noncytopathic clearance”. This sentence was not accurate and lack of references.

Response: Your comments are truly appreciated. The sentence is now revised from “During the early stage of HBV infection, HBV replication is suppressed by innate immunity through antiviral cytokines without damage to infected hepatocytes, a phenomenon dubbed noncytopathic clearance.” to “HBV clearance during acute HBV infection is mainly mediated by antiviral cytokines secreted by cytotoxic T lymphocytes without damage to infected hepatocytes, a phenomenon dubbed noncytopathic clearance (Guidotti et al., 1994, 1999; McClary et al., 2000).”

6. Which method was used for IL-32 γ detection in supernatant? The unit of IL-32 γ used in Fig 1F,G and other figures seems not correct.

Response: Thank you for your remarks. As described in the Fig. 1f legend, the total amount of secreted (supernatant) or intracellular IL-32 from cells grown in one well (5×10^5 cells) of a 12-well plate is shown on the Y-axis. The Y-axis units in Fig. 1 and 2 have been revised to "IL-32 (ng/ 5×10^5 cells). For clarity, we newly added a detailed explanation of IL-32 ELISA in the Materials and Methods section.

7. The quality of HBV core protein western blot in Fig. 3B is bad and please repeat the experiment.

Response: Following the reviewer's comment, we repeated the Fig. 3b experiment and determined the level of the HBV core protein by Western blotting. The results are included in revised Fig. 3b.

Fig. 3b

8. Line 319-320. "A literature search..... EnhII/Cp". This sentence was not accurate and lack of references. As reviewed by Quasdorf M. et al (J Viral Hepat. 2010 Aug;17(8):527-36), there are more TFs than these four genes.

Response: Thank you for your insight. The sentence is now revised from "A literature search revealed that, among liver-enriched transcription factors, only C/EBP, HNF1, HNF3, and HNF4 bind to both EnhI and EnhII/Cp (Fig. 4a)." to "A literature search revealed that several ubiquitous and liver-enriched transcription factors bind to both EnhI and EnhII/Cp

(Quasdorff and Protzer, 2010; Kim et al., 2016). Among those, we focused on the major liver-enriched transcription factors known to be involved in HBV transcription including C/EBP, HNF1, HNF3, and HNF4 (Fig. 4a).”

9. line 385-386: “Suppression of HBV by cytokines.....” The figure was missing.

Response: Thank you for your careful remarks. We included the corresponding Figure in revised manuscript: “Suppression of HBV by cytokines.....(Supplementary Fig. 11)”.

Reviewer #2: Kim et al. report here that interleukin (IL)-32 is an intracellular downstream mediator involved in the noncytopathic inhibition of HBV induced by IFN- γ and TNF- α . The Authors build on a growing body of literature that shows that IL-32 inhibits the replication of several viruses, including VSV, HIV and influenza. Kim et al. claim that the mechanism of action whereby IL-32 inhibits HBV occurs at the transcriptional level, and involves a decreased viral enhancer activity by ERK1/2-dependent down-regulation of liver-enriched transcription factors. Although potentially of interest, there are a number of conceptual and technical issues that reduce my enthusiasm for the work. Specifically:

1) The claim that the IL-32-mediated inhibition of HBV occurs at the transcriptional level is at odds with previously published literature (the Authors are referred to Guidotti and Chisari, Annual Review of Pathology 2006 for a comprehensive review of those studies). Briefly, the noncytopathic inhibition of HBV replication by IFN- γ was shown to occur at the post-transcriptional level, with minimal or no effect on HBV transcription. IFN- γ was shown to prevent the assembly of replication-competent HBV RNA-containing capsids in the hepatocyte in a proteasome- and kinase-dependent manner. The viral nucleocapsids disappear from the cytoplasm of the hepatocytes, and the viral RNAs are destabilized by a SSB/La-dependent mechanism in the nucleus. How can the Authors reconcile such discrepancies? Have they tested a potential post-transcriptional activity of IL-32?

Response: We appreciate this important comment. The pioneering work of Dr. Guidotti and Dr. Chisari established that IFN- γ inhibits HBV at the post-transcriptional level in a mouse model (Guidotti and Chisari, 2006). Using HBV TG mice and stable mouse cells, they showed that the anti-viral effect of cytokines occurs at the post-transcriptional rather than transcriptional level: 1) The stability of HBV RNA is reduced by cytokine-induced depletion of the RNA-binding La-protein (Tsui et al., 1995; Heise et al., 1999a; Heise et al., 1999b). 2) HBV RNA-containing capsids are disappeared via proteasome (Robek et al., 2002; Wieland et al., 2005) or kinase activity (Robek et al., 2004).

Therefore, we have carefully examined whether there is any difference between mouse and human hepatocytes in terms of cytokine effect on HBV suppression.

Since our results supported that the cytokine-induced IL-32 inhibits HBV at the transcriptional level, we first determined the effect of IL-32 knock-down on cytokine-induced suppression of HBV by Northern blot analysis (Fig. 3e). The cytokine-induced decrease in

HBV RNAs was significantly reverted by IL-32 knock-down in human hepatocytes (Fig. 3e). Together with IL-32-mediated inhibition of enhancer activity (Fig. 3d), our data suggest that cytokines inhibit HBV mainly at the transcriptional level inducing IL-32 in Huh7 cells. These results are consistent with our and other previous studies which found that cytokines inhibit HBV at the transcriptional level in human hepatocytes (Mao R, et al., 2011; Park et al., 2016; Lim et al., 2018).

Fig. 3e

We also analyzed the effect of cytokines on HBV replication and antigen expression in primary mouse hepatocytes (PMHs). We freshly prepared PMHs from mouse liver, transfected them with the HBV 1.2 plasmid and treated them with cytokines. The level of viral RNA and antigen secretion were decreased by cytokine treatment and IL-32 γ expression in PMHs (Supplementary Fig. 7).

Next, we checked whether the SSB/La protein level is affected by cytokine treatment or IL-32 γ expression because the stability of HBV RNA is reduced by cytokine-induced depletion of the La protein in a mouse model (Tsui et al., 1995; Heise et al., 1999a; Heise et al., 1999b). Indeed, cytokines strongly destabilized the La protein in PMHs, like in previous studies, however they showed little effect in Huh7 human cells (Supplementary Fig. 7). Importantly, the level of the La protein was not affected by overexpression of IL-32 γ in either cell type, suggesting that cytokine-induced IL-32 downregulates HBV RNA in a La-independent manner.

In addition, as shown in Supplementary Fig. 7d, IL-32 is absent in mice (please refer to our response to Reviewer #1's comment 4).

Supplementary Fig. 7

To further confirm the La-independence in human hepatocytes, we determined the levels of HBV RNAs after a knock-down of La expression (Supplementary Fig. 8a). In Huh7 cells, La knock-down had no effect on the levels of HBV RNAs, whereas IL-32 strongly reduced them (Supplementary Figs. 8b and 9a). However, La knock-down in PMHs significantly decreased the levels of HBV RNAs and antigens (Supplementary Fig. 9b). These data demonstrate that cytokines suppress HBV RNAs at the post-transcriptional level through a La-dependent pathway in PMHs but at the transcriptional level through a La-independent pathway in human hepatocytes.

Supplementary Fig. 8

Supplementary Fig. 9

Finally, we tested the effect of proteasome and kinase activity on IL-32-mediated

downregulation of HBV RNA because previous studies demonstrated that cytokines destabilize the RNA-containing nucleocapsids via proteasome (Robek et al., 2002; Wieland et al., 2005) or kinase activity (Robek et al., 2004) in mouse hepatocytes. However, there was no restoration of IL-32-mediated downregulation of HBV RNA or antigens when we treated Huh7 cells with the same inhibitors of JAK (AG490) and proteasome (MG132) that were used in previous studies (Supplementary Fig. 14). These results demonstrate that the anti-viral effect of IL-32 is independent of proteasome or kinase activity, at least in human hepatocytes.

Altogether, our results suggest that cytokines suppress HBV at the post-transcriptional level in mouse hepatocytes, mainly through a La-dependent pathway; however, they suppress HBV at the transcriptional level in human hepatocytes through an IL-32-dependent pathway. The difference in the mechanism involved in cytokine-mediated suppression of HBV between human and mouse is probably due to the fact that mouse does not have any IL-32 homologs.

Supplementary Fig. 14

2) If the HBV-inhibiting activity of IL-32 is exclusively intracellular, how do the Authors

explain the results reported in Fig. 6? The in vivo transduction efficiency of the hydrodynamic injection of a plasmid is known not to exceed 20% of the hepatocytes. Are we to assume that both HBV and IL-32 (encoded by two separate plasmids) randomly transduced the exact same hepatocytes?

Response: Your comments are truly appreciated. According to our results, IL-32 strongly suppresses HBV, although transduction efficiency of IL-32 by hydrodynamic injection is low in mouse liver (Fig. 6). Since we showed that IL-32-mediated inhibition of HBV replication is an intracellular event, if there is no paracrine effect, the IL-32 and HBV plasmids transduced by hydrodynamic injection should be mostly co-localized in same hepatocytes.

To check this, we transduced a mixture of RFP and GFP plasmids (1:1) into mouse liver and analyzed whether the two proteins are expressed in the same hepatocytes (Fig. 6d). Indeed, confocal microscopic analysis showed that RFP and GFP are co-expressed in most of the transduced hepatocytes. Our observation is consistent with the previous reports that the two plasmids delivered by hydrodynamic injection were co-localized in 91% of hepatocytes examined (Sebestyen et al., 2006; Zhu et al., 2006).

In addition, when a mixture of HBV and β -gal plasmids (with or without IL-32 γ) was hydrodynamically injected into mouse liver, confocal microscopic analysis after immunostaining showed that the HBV core protein and β -gal co-localized in most of the transduced hepatocytes (Fig. 6c). Although the expression of the core protein was suppressed by IL-32, it seems evident that two proteins were co-localized in same hepatocytes. These results suggest that IL-32-mediated inhibition of HBV in mouse liver is due to the co-expression of IL-32 and HBV in the same hepatocytes, not to the paracrine effect of IL-32. These results are described in the revised text.

Fig. 6c and 6d

3) In Fig. 2E-F, the Authors claim that IL-32 does not have any effect on the replication of other viruses, such as HSV-1 or influenza. To make such a claim, the Authors should select viruses and design experimental conditions whereby the same combination of TNF- α and IFN- γ shown to inhibit HBV in Fig. 1 inhibits replication of said viruses as well.

Response: Following this comment, we performed additional experiments using HSV and IAV with treatment with different concentrations of TNF- α and IFN- γ that inhibit HBV replication.

We found that HSV-1 is insensitive to a combination of TNF- α and IFN- γ ; however, influenza virus was sensitive to these cytokines (Fig. 2e-h). Still, IL-32 showed no effect on replication of either of these viruses. Therefore, we have added the following sentences: "First, we found that HSV-1 is insensitive to a combination of TNF- α and IFN- γ , which inhibits HBV replication; however, influenza virus was sensitive to these cytokines (Fig. 2e and 2f). Analysis of GFP signals and viral titration showed that IL-32 γ expression did not affect the replication of HSV-1 or influenza virus in Huh7 cells (Fig. 2e-h)".

We also analyzed the levels of viral proteins to confirm the sensitivity of each virus to the cytokine combination (Supplementary Figs. 4 and 5c-d).

Figure 2

Supplementary Fig. 4

Supplementary Fig. 5c and d

4) In the Introduction, the Authors state: “During the early stage of HBV infection, HBV replication is suppressed by innate immunity through antiviral cytokines without damage to infected hepatocytes, a phenomenon dubbed noncytopathic clearance.” The alleged role of innate immunity in viral clearance during natural HBV infection is unsubstantiated and contrasts our current understanding of the immune response

against HBV.

Response: We truly appreciate this comment and have revised the sentence to “HBV clearance during acute HBV infection is mainly mediated by antiviral cytokines secreted by cytotoxic T lymphocytes without damage to infected hepatocytes, a phenomenon dubbed noncytopathic clearance (Guidotti et al., 1994, 1999; McClary et al., 2000).”

5) The Authors are encouraged to provide comprehensive methodological details to allow a thorough understanding of the experiments. In many instances, I could not find such information in the Results, Methods or Figure legend section.

Response: Thank you for your remarks. We have strengthened the methodological details in the revised manuscript, mainly in the Methods section.

Minor:

1) Can the Authors provide a positive control for the anti-IL-32 blocking antibody (e.g. by showing that it effectively blocks IL-32 activity in the assay reported in Supplementary Fig. 2)?

Response: Thank you for this insightful comment. We asked the antibody manufacturer about the neutralizing activity of the IL-32 antibody. The manufacturer told us that the IL-32 antibody has been used for western blotting and ELISA rather than as a neutralizing antibody. Therefore, we removed the figure in revised manuscript. We also added specific information on IL-32 antibody, including the catalog number, in the Materials and Methods section to avoid confusion.

2) Does IL-32 induce TNF- α and IFN- γ ?

Response: According to the reviewer’s comment, we investigated the effect of IL-32 on cytokine induction in PHHs. Our data show that IL-32 does not induce TNF- α or IFN- γ in PHHs (Supplementary Fig. 6b).

Supplementary Fig. 6b

3) Why do the mice injected with the IL-32 plasmids stain positive for b-gal?

Response: As described in the Fig. 6 legend, the β -gal plasmid along with HBV and IL-32 plasmids was hydrodynamically injected into mouse liver as an expression control.

References

Dorner T, Hucko M, Mayet WJ, Trefzer U, Burmester GR, Hiepe F. Enhanced membrane expression of the 52 kDa Ro(SS-A) and La(SS-B) antigens by human keratinocytes induced by TNF alpha. *Ann Rheum Dis* 54, 904-909 (1995).

Guidotti LG, et al. Cytotoxic T lymphocytes inhibit hepatitis B virus gene expression by a noncytolytic mechanism in transgenic mice. *Proc Natl Acad Sci U S A* 91, 3764-3768 (1994).

Guidotti LG, Rochford R, Chung J, Shapiro M, Purcell R, Chisari FV. Viral clearance without destruction of infected cells during acute HBV infection. *Science* 284, 825-829 (1999).

Guidotti LG, Chisari FV. Immunobiology and pathogenesis of viral hepatitis. *Annu Rev Pathol* 1, 23-61 (2006).

Heise T, Guidotti LG, Cavanaugh VJ, Chisari FV. Hepatitis B virus RNA-binding proteins associated with cytokine-induced clearance of viral RNA from the liver of transgenic mice. *J Virol* 73, 474-481 (1999a).

Heise T, Guidotti LG, Chisari FV. La autoantigen specifically recognizes a predicted stem-loop in hepatitis B virus RNA. *J Virol* 73, 5767-5776 (1999b).

Kim DH, Kang HS, Kim KH. Roles of hepatocyte nuclear factors in hepatitis B virus infection. *World J Gastroenterol* 22, 7017-7029 (2016).

Kim SH, Han SY, Azam T, Yoon DY, Dinarello CA. Interleukin-32: a cytokine and inducer of TNF α . *Immunity* 22, 131-142 (2005).

Li Y, et al. Inducible Interleukin 32 (IL-32) Exerts Extensive Antiviral Function via Selective Stimulation of Interferon lambda1 (IFN-lambda1). *J Biol Chem* 288, 20927-20941 (2013).

Lim KH, et al. Suppression of interferon-mediated anti-HBV response by single CpG methylation in the 5'-UTR of TRIM22. *Gut* 67, 166-178 (2018).

Mao R, et al. Indoleamine 2,3-dioxygenase mediates the antiviral effect of gamma interferon against hepatitis B virus in human hepatocyte-derived cells. *J Virol* 85, 1048-1057 (2011).

McClary H, Koch R, Chisari FV, Guidotti LG. Relative sensitivity of hepatitis B virus and other hepatotropic viruses to the antiviral effects of cytokines. *J Virol* 74, 2255-2264 (2000).

Park YK, et al. Cleaved c-FLIP mediates the antiviral effect of TNF- α against hepatitis B virus by dysregulating hepatocyte nuclear factors. *J Hepatol* 64, 268-277 (2016).

Quasdorff M, Protzer U. Control of hepatitis B virus at the level of transcription. *J Viral Hepat* 17, 527-536 (2010).

Robek MD, Wieland SF, Chisari FV. Inhibition of hepatitis B virus replication by interferon requires proteasome activity. *J Virol* 76, 3570-3574 (2002).

Robek MD, Boyd BS, Wieland SF, Chisari FV. Signal transduction pathways that inhibit hepatitis B virus replication. *Proc Natl Acad Sci U S A* 101, 1743-1747 (2004).

Sebestyen MG, et al. Mechanism of plasmid delivery by hydrodynamic tail vein injection. I. Hepatocyte uptake of various molecules. *J Gene Med* 8, 852-873 (2006).

Tsui LV, Guidotti LG, Ishikawa T, Chisari FV. Posttranscriptional clearance of hepatitis B virus RNA by cytotoxic T lymphocyte-activated hepatocytes. *Proc Natl Acad Sci U S A* 92, 12398-12402 (1995).

Wieland SF, Eustaquio A, Whitten-Bauer C, Boyd B, Chisari FV. Interferon prevents formation of replication-competent hepatitis B virus RNA-containing nucleocapsids. *Proc Natl Acad Sci U S A* 102, 9913-9917 (2005).

Zhu HZ, Wang W, Feng DM, Sai Y, Xue JL. Conditional gene modification in mouse liver using hydrodynamic delivery of plasmid DNA encoding Cre recombinase. FEBS Lett 580, 4346-4352 (2006).

Reviewers' comments:

Reviewer #1 (Remarks to the Author):

My comments on this manuscript were properly answered and I have no further question.

Reviewer #2 (Remarks to the Author):

The revised study still falls short of convincingly showing that the alleged IL-32 mediated inhibition of HBV occurs at the transcriptional level. This has to be done the hard way, i.e. by performing run-on/run-off experiments as well as by RNA analyses from nuclear and cytoplasmic fractions of cytokine-treated hepatocytes. In the absence of such experiments, it is impossible to claim that the reduction of total HBV RNA in human hepatocytes reflects transcriptional impairment of HBV genes.

The experiments performed in mice are uninterpretable as the human cytokine was used and data presented in the rebuttal suggest different mechanisms of action of IFN-g and TNF-a in human versus mouse hepatocytes. Moreover, the data provided to claim a very high percentage of hepatocyte co-transduction upon hydrodynamic injection of two different plasmids are not convincing (lower magnification images should be provided, and a rigorous quantification performed).

These considerations, coupled with the observation that an alleged antiviral role for IL-32 against HBV has already been proposed, lower my enthusiasm for the study.

Response to reviewer' comments

Reviewer #1 (Remarks to the Author):

My comments on this manuscript were properly answered and I have no further question.

Response: Thank you for your helpful comments during revision.

Reviewer #2 (Remarks to the Author):

1) The revised study still falls short of convincingly showing that the alleged IL-32 mediated inhibition of HBV occurs at the transcriptional level. This has to be done the hard way, i.e. by performing run-on/run-off experiments as well as by RNA analyses from nuclear and cytoplasmic fractions of cytokine-treated hepatocytes. In the absence of such experiments, it is impossible to claim that the reduction of total HBV RNA in human hepatocytes reflects transcriptional impairment of HBV genes.

Response: Thank you for these suggestions. To confirm that the IL-32-mediated inhibition of HBV occurs at the transcriptional level, we newly performed a nuclear run-on assay (Fig. 3g) as well as Northern blot analysis (Fig. 3f) of nuclear and cytoplasmic fractions of IL-32-transfected or IFN- γ -treated human hepatocytes. The steady-state level of HBV transcription was suppressed by both IL-32 γ and IFN- γ in the nuclear and cytoplasmic compartments (Fig. 3f). Furthermore, nuclear run-on assay revealed that the rate of HBV transcription was decreased by IL-32 γ and IFN- γ (Fig. 3g). Together with IL-32-mediated inhibition of enhancer activity (Fig. 3d), our data suggest that cytokines inhibit HBV at the transcriptional level by inducing IL-32 in human hepatocytes.

2) The experiments performed in mice are uninterpretable as the human cytokine was used and data presented in the rebuttal suggest different mechanisms of action of IFN- γ and TNF- α in human versus mouse hepatocytes.

Response: We did not use human cytokines in mouse primary hepatocytes. Mouse hepatocytes were treated with mouse cytokines (denoted as mTNF- α and mIFN- γ) and human hepatocytes were treated with human cytokines (denoted as TNF- α and IFN- γ) throughout the study.

3) Moreover, the data provided to claim a very high percentage of hepatocyte co-transduction upon hydrodynamic injection of two different plasmids are not convincing (lower magnification images should be provided, and a rigorous quantification performed). These considerations, coupled with the observation that an

alleged antiviral role for IL-32 against HBV has already been proposed, lower my enthusiasm for the study.

Response: Thank you for commenting. We newly performed experiments to ascertain that the two plasmids transduced by hydrodynamic injection were mostly co-localized in the same hepatocytes and added low-magnification ($\times 100$) images (Fig. 6d). Confocal microscopic analysis revealed that RFP and GFP were co-expressed in most of the transduced hepatocytes. Instead of quantification, we only provide raw confocal images because the extent of co-localization was slightly different between cells. Our observation is consistent with the previous reports that the two plasmids delivered by hydrodynamic injection were co-localized in 91% of hepatocytes examined (Sebestyen et al., 2006; Zhu et al., 2006).

Li et al. (2013) suggested that treatment of peripheral blood mononuclear cells with recombinant IL-32 induced IFN- $\lambda 1$, which in turn inhibited HBV replication in hepatocytes indirectly. However, the direct antiviral action of IL-32 in hepatocytes has not been demonstrated. In this study, we show that intracellular IL-32 γ is a downstream mediator of the anti-HBV activity of TNF- α and IFN- γ . IL-32 γ inhibits HBV at the transcriptional level in human hepatocytes and functions as a non-cytokine-like molecule in HBV suppression.

REVIEWERS' COMMENTS:

Reviewer #2 (Remarks to the Author):

The Authors have addressed my first concern and now show that cytokines such as IFN-g and TNF-a suppress HBV RNA at the transcriptional level in humans (via IL-32).

However, the experiments performed in mice are uninterpretable as the human IL-32 was used (mice do not have IL-32) and the data presented in the paper suggest different mechanisms of action of IFN-g and TNF-a in human versus mouse hepatocytes. As such, I think the results obtained in mice should be removed and the paper should focus instead on the proposed IL-32 mediated inhibition of HBV transcription in human hepatocytes.

Response to reviewer's comment:

Reviewer #2 (Remarks to the Author):

The Authors have addressed my first concern and now show that cytokines such as IFN-g and TNF-a suppress HBV RNA at the transcriptional level in humans (via IL-32).

However, the experiments performed in mice are uninterpretable as the human IL-32 was used (mice do not have IL-32) and the data presented in the paper suggest different mechanisms of action of IFN-g and TNF-a in human versus mouse hepatocytes. As such, I think the results obtained in mice should be removed and the paper should focus instead on the proposed IL-32 mediated inhibition of HBV transcription in human hepatocytes.

→ Thank you for your comment. We think that providing the information about the existence of different mechanisms of action of IFN-g and TNF-a in human versus mouse hepatocytes to general readers is important to help their understanding the nature of HBV immunology. Therefore, we want to provide all the data obtained from mouse and human hepatocytes.